# Emerging Gene Therapy Based on Nanocarriers: A Promising Therapeutic Alternative for Cardiovascular Diseases and a Novel Strategy in Valvular Heart Disease

**DOI:** 10.3390/ijms26041743

**Published:** 2025-02-18

**Authors:** Haoran Yang, Junli Li, Chengxiang Song, Hongde Li, Qiang Luo, Mao Chen

**Affiliations:** 1Laboratory of Cardiac Structure and Function, Institute of Cardiovascular Diseases, West China Hospital, Sichuan University, Chengdu 610041, China; haoranyang@stu.scu.edu.cn (H.Y.); ljl@wchscu.cn (J.L.); songchengxiang1994@163.com (C.S.);; 2Department of Cardiology, West China Hospital, Sichuan University, No.37 Guoxue Street, Chengdu 610041, China; 3Cardiac Structure and Function Research Key Laboratory of Sichuan Province, West China Hospital, Sichuan University, Chengdu 610041, China

**Keywords:** gene therapy, nanocarriers, coronary heart disease, pulmonary hypertension, hypertension, valvular heart disease

## Abstract

Cardiovascular disease remains a leading cause of global mortality, with many unresolved issues in current clinical treatment strategies despite years of extensive research. Due to the great progress in nanotechnology and gene therapy in recent years, the emerging gene therapy based on nanocarriers has provided a promising therapeutic alternative for cardiovascular diseases. This review outlines the status of nanocarriers as vectors in gene therapy for cardiovascular diseases, including coronary heart disease, pulmonary hypertension, hypertension, and valvular heart disease. It discusses challenges and future prospects, aiming to support emerging clinical treatments. This review is the first to summarize gene therapy using nanocarriers for valvular heart disease, highlighting their potential in targeting challenging tissues.

## 1. Introduction

Cardiovascular disease is a leading cause of death globally [1]. In 2019, cardiovascular disease was responsible for about one-third of all fatalities, affecting 9.6 million men and 8.9 million women [1]. Currently, the clinical management of cardiovascular disease predominantly depends on pharmacological and surgical interventions. Despite extensive efforts over the years, there are still numerous unsolved problems. For instance, pharmacological interventions for coronary heart disease (CHD) are ineffective in preventing atherosclerotic plaque progress in some cases. Surgical treatments for CHD are suboptimal, as percutaneous coronary intervention (PCI) involves risks like stent thrombosis and restenosis, and coronary artery bypass grafting (CABG) is associated with substantial trauma and increased postoperative complications [2]. When it comes to hypertension, many oral formulations of antihypertensive medications demonstrate limited water solubility and undergo significant first-pass metabolism, leading to diminished systemic bioavailability [3]. Consequently, higher dosages are required to achieve the intended therapeutic outcomes, which may concurrently increase the risk of adverse side effects [3]. In addition, the heterogeneity of the etiology may also contribute to the low successful treatment rates [4]. And in certain rare diseases such as pulmonary artery hypertension, the range of available drugs remains limited, and the prognosis of these patients is still very poor [5,6,7,8].

Consequently, it is imperative to investigate new therapeutic approaches which are effective for cardiovascular disease. Research has shown that changes in gene expression are closely associated with the development of numerous cardiovascular diseases [4,9,10,11]. Nevertheless, molecular targeted therapies that address these pathogenic mechanisms remain in the early stage of investigation and warrant further attention. Gene therapy refers to the incorporation of exogenous genes into target cells utilizing gene transfer technologies, with the aim of rectifying diseases caused by genetic abnormalities and thereby achieving therapeutic outcomes. In recent years, gene therapy has experienced rapid progress. Since 2016, six gene therapy products have been approved by both the European Medicines Agency (EMA) and the United States Food and Drug Administration (FDA). Currently, there are over 800 cell and gene therapy programs undergoing clinical development, targeting diseases that were previously considered untreatable [12]. Recently, gene therapy has rapidly expanded in cardiovascular medicine, providing a novel treatment approach for cardiovascular diseases. For example, the administration of adeno-associated virus (AAV) to deliver plakophilin 2 (PKP2) in PKP2c.2013delC/WT mice effectively restored the levels of cardiac PKP2 as well as other junctional proteins and enhanced cardiac function, which indicated its potential as a therapeutic strategy for arrhythmogenic cardiomyopathy [13].

The methods for delivering gene therapy include physical strategies (such as electroporation, gene guns, fluid pressure, sonoporation, and magnetic transfection) and vector-based delivery. Vectors in gene therapy are classified as either viral or non-viral. The main viral vectors include adenoviruses, adeno-associated viruses, retroviruses, and lentiviruses. In contrast, non-viral vectors primarily include liposomes, cell-penetrating peptides, and nanoparticles [14]. Compared to viral vectors, nanocarriers not only mitigate potential biosecurity risks and allow for chemical modification but also possess attributes conducive to controlled drug delivery [15] (such as pH-, ROS-, or enzyme-responsive nanoparticles [16,17,18]). In addition, the range of materials available for the formulation of nanocarriers is extensive, encompassing lipids, metals, polymers, and inorganic compounds, among others [19]. Based on the above advantages, the potential and value of nanocarriers are increasingly being recognized, garnering significant attention. In the realm of cardiovascular disease, nanocarrier-based gene therapy is progressively advancing towards clinical practice [20]. The gene therapy VERVE-101 targets and inactivates the proprotein convertase subtilisin/kexin type 9 (PCSK9) gene in the liver to lower low-density lipoprotein cholesterol (LDL-C) levels, showing potential in reducing cardiovascular disease risk, as demonstrated by animal and ongoing clinical trials [20].

This review comprehensively summarizes the status of nanocarriers as vectors for gene therapy in treating cardiovascular diseases, focusing on coronary heart disease, pulmonary hypertension, hypertension, and valvular heart disease. Additionally, we will address the associated challenges and future prospects, intending to provide a foundational basis and support for these emerging clinical treatment methodologies. This review uniquely consolidates information on nanocarrier-based gene therapy for valvular heart disease. (Examples of gene therapy utilizing nanocarriers in the context of cardiovascular diseases are presented in Figure 1, and a concise overview of the process in which nanocarriers enter the cell is provided in Figure 2. In addition, Table 1 provides a comprehensive list of all nanocarrier-based studies reviewed in this paper).

## 2. Categories of Nanocarriers

### 2.1. Inorganic Nanocarriers

Inorganic nanocarriers are classified by composition into metallic nanoparticles, like gold and titanium dioxide, and nonmetallic nanoparticles, such as carbon and mesoporous silica. Mesoporous silica nanoparticles (MSNs) have recently gained recognition as promising candidates for biopharmaceutical applications. MSNs are distinguished by their stable structure, extensive surface chemistry, and high dispersibility [62]. Moreover, MSNs exhibit tunable pore sizes and versatile drug loading and release properties, establishing them as a promising innovation in drug delivery systems [15]. Additionally, the surfaces of MSNs can be functionalized with specific ligands to target lesion sites. For example, a PEGylated porous silicon nanocarrier functionalized with atrial natriuretic peptide (ANP) was developed by Ferreira et al. to target receptors overexpressed in the ischemic endocardium, enhancing the delivery of cardioprotective drugs to the infarcted area [63]. These nanoparticles provide enhanced flexibility, versatility, and robustness compared to traditional drug delivery systems like polymer-based and lipid-based nanocarriers [62]. The customizable pore size and geometry of their long-range ordered structure enable uniform incorporation of guest molecules with diverse sizes and properties. Furthermore, MSNs offer excellent monodispersity and dispersibility, with customizable size and structure, ensuring consistent in vivo pharmacokinetics and predictable outcomes [62].

Hollow/rattle-type mesoporous silica nanoparticles (MSNs) can be synthesized using three main methods: the soft template method, the selective etching strategy, and the self-template method [62]. Schematic illustrations of the three methods are presented in Figure 3A. The soft template method for synthesizing hollow or rattle-type MSNs employs surfactants as templates, often using dual or multi-surfactants to form a complex structure that creates both a mesoporous shell and a hollow core [62]. Although this method is simple and effective, it has limitations, including challenges in controlling the particle size and shell thickness over a broad range, difficulties in scaling up production, and incomplete removal of the template surfactant [62]. The selective etching strategy has been enhanced using various organosilane precursors. This approach employs a pure silica framework combined with hybrid organic–inorganic networks to create various compositions and structures, demonstrating different stability levels when subjected to distinct etching agents or specific temperature or pH conditions. This method is notable for being simple, effective, scalable, controllable, and cost-efficient. Additionally, the particle size, core diameter, and shell thickness can be extensively modified [62]. The self-template method synthesizes hollow mesoporous silica nanoparticles without requiring an external template. This approach is both simple and cost-effective; however, challenges remain in scaling up the reaction process while maintaining stability and dispersibility [62].

### 2.2. Polymer-Based Nanocarriers

Polymeric nanocarriers, comprising nanocapsules and nanospheres, are synthesized from polymers and typically measure between 1 and 1000 nm in size. These nanoparticles have gained prominence as drug delivery systems due to their good biodegradability, bioavailability, and controllability in size, shape, and surface charge, as well as their ability to facilitate controlled drug release [64,65]. Polymer nanoparticles made from polylactic acid (PLA), polyglycolic acid (PGA), and poly(lactic-co-glycolic acid) (PLGA) are widely used for delivering therapeutic agents like drugs, DNA, and proteins, and have been approved by the United States Food and Drug Administration (FDA) and the European Medicines Agency (EMA) [66]. Due to its low solubility and rapid degradation into glycolic acid, poly(glycolic acid) (PGA) is considered suboptimal for drug delivery applications [67]. Poly(lactic acid) (PLA) and poly(lactic-co-glycolic acid) (PLGA) are widely used in drug delivery systems due to their biodegradability and adjustable mechanical properties [68]. These polymers can be synthesized with varying molecular weights and lactic-to-glycolic acid (L:G) ratios, allowing for customization to specific applications with high reproducibility and cost-effectiveness [68]. PLGA, in particular, is characterized by minimal systemic toxicity and is one of the most successfully employed biodegradable polymers [69]. Its hydrolysis yields lactic acid and glycolic acid, which are endogenous metabolites readily processed by the body. Conversely, PLA has been utilized to a lesser extent than PLGA, primarily due to its slower degradation rate [69]. In addition, rather than serving as a carrier, polyethylene glycol (PEG), another member in this family, is commonly conjugated to the surfaces of various nanocarriers to mitigate opsonization, enhance circulation time, and improve accumulation at the target site [70].

Dendritic macromolecules, a subtype within this family, possess a dendritic structure consisting of an initiator core, branches extending from this core, and functional termini at the branch ends. The polyamidoamine (PAMAM) dendrimer represents the first and most extensively utilized synthesized molecule of this type. PAMAM’s distinctive dendritic structure, marked by exponential growth and high terminal functional density, sets it apart from other polymers. Additionally, its substantial internal cavities render it an ideal candidate for the encapsulation and adsorption of biomolecules [71,72,73].

The preparation methods for polymer-based nanocarriers encompass self-assembly, nanoprecipitation, dialysis, emulsion-based self-assembly, emulsion polymerization, ion gelation/sol–gel, polymerization-induced self-assembly, spray drying, and templated assembly techniques [65]. Self-assembly, the spontaneous organization of distinct polymer chains into a highly controlled particle suspension, is the predominant method for synthesizing polymeric nanoparticles [65]. This process entails a complex interaction between the system’s internal energy (enthalpy), encompassing covalent and supramolecular forces like electrostatic interactions, hydrogen bonding, and van der Waals forces, and the system’s entropy [74]. For instance, during the self-assembly of amphiphilic polymers in an aqueous environment, water molecules at the interface between the hydrophobic polymer and the bulk solution are constrained into a unique, well-ordered state with significantly reduced degrees of freedom, characterizing a low-entropy system [75]. Consequently, nanoparticles are formed through an entropically favorable process to minimize these surface interactions. Figure 3B illustrates the synthesis of PLGA-based nanocarriers via the self-assembly method.

### 2.3. Lipid-Based Nanocarriers

Lipid-based nanocarriers exhibit biocompatibility and high bioavailability, capable of encapsulating both hydrophilic and hydrophobic compounds. Their varied physicochemical properties can be precisely adjusted to affect biological behavior. Their simplicity, scalability, and cost-effective production make these attributes highly beneficial for targeted therapeutic applications [76]. Liposomes and lipid micelles are frequently utilized lipid-based nanocarriers. The ability of lipids to form micelles or liposomes is mainly influenced by the size ratio between their hydrophobic tails and hydrophilic headgroups. When the headgroup is much wider than the tail, the molecule takes on a cone-like shape, resulting in micelle formation. When the headgroup and tail have similar widths, the molecule takes on a cylindrical shape, facilitating liposome formation [77,78].

Lipid nanoparticles (LNPs), particularly cationic LNPs, represent the most prevalent non-viral delivery system for nucleic acid drugs [79,80]. LNPs utilize synthetic cationic lipids to create stable complexes with anionic nucleic acids, enhancing their stability and resistance to nuclease degradation, thus facilitating in vivo delivery [81]. LNPs enhance the therapeutic efficacy of nucleic acid drugs by facilitating endosomal escape, cellular uptake, and drug release, enabling them to cross biological barriers [82]. However, there are notable limitations associated with LNPs. Administration beyond a specific dosage threshold of LNPs containing ionizable lipids may lead to immunotoxicity. The activation of the innate immune response occurs when phagocytic cells in the reticuloendothelial system (RES) detect the lipid components of LNPs. This activation can lead to the stimulation of toll-like surface receptors, resulting in the induction of elevated cytokine levels, a condition known as cytokine release syndrome [83,84]. LNPs can activate serum complement, causing non-IgE-mediated hypersensitivity reactions known as complement activation-related pseudoallergy, which may result in anaphylactic shock [85,86,87].

LNP systems, consisting of ionizable cationic lipids, helper lipids, and nucleic acid polymers, are synthesized using various mixing techniques [88]. The synthesis processes are illustrated in Figure 3C. These methods rely on the initial creation of small, positively charged vesicles that form complexes with nucleic acid polymers, including small interfering RNA (siRNA), messenger RNA (mRNA), and plasmid DNA (pDNA), in a pH 4 buffer [89]. Lipoplexes are structured systems where lipid–nucleic acid complexes form configurations like multilamellar structures, positioning the nucleic acid cargo between stacked bilayers [90]. Aggregation is inhibited by incorporating a minor proportion of PEG lipid into the lipid mixture. Dialysis against phosphate-buffered saline (PBS) raises the pH to 7.4, causing the ionizable cationic lipid to become neutral and phase-separate, forming an amorphous oil droplet inside the LNP [89]. The morphology of the LNP is affected by the choice of ionizable cationic lipid and the method of fusion induction, either through pH elevation or formulation in a high-ionic-strength medium at pH 4, resulting in bleb structures [91].

### 2.4. Microbubbles

Microbubbles are composed of a gas core characterized by low solubility in water, encapsulated by a shell made of macromolecular material. This configuration causes an acoustic impedance mismatch with biological fluids and tissues, making microbubbles ideal for ultrasound imaging and potential cardiovascular drug delivery applications [92]. A promising therapeutic approach utilizes ultrasound-targeted microbubble destruction (UTMD) to exploit the cavitation effect of microbubbles [93,94,95]. Gene drugs, including pDNA and siRNA, demonstrate significant efficacy when administered at low dosages in conjunction with a limited quantity of microbubbles [92]. For instance, cationic microbubbles (CMBs) exhibit a robust capacity for DNA binding attributable to their positive charge, which protects negatively charged DNA from degradation in the bloodstream, thereby enhancing gene transfection efficiency [96,97]. Although the transfection efficacy of microbubbles is relatively low compared to viral vectors, they can achieve selective nucleic acid delivery in ultrasound-treated regions [98]. Additionally, microbubbles can be mass-produced, frozen, and stored for a long period [99], which represents a significant advantage.

Over the past few decades, advancements in microbubble production methods have enhanced their efficacy, and the processes of synthesis are depicted in Figure 3D. Microbubbles can be generated by mixing phospholipids and surfactants in a test tube, resulting in heterogeneous micelles that can be stored and loaded with drugs as needed [100,101]. More sophisticated techniques, such as microfluidics, facilitate the production of microbubbles with uniform size by allowing precise control over gas and inlet pressures, thereby yielding homogeneous microbubbles [101,102]. Following production, microbubbles are subjected to centrifugation, washing, and purification, enabling labeling with specific markers for targeted applications [103,104].

### 2.5. Exosomes

Exosomes, 50–150 nm vesicles formed by plasma membrane invagination, enable intercellular transport of proteins, nucleic acids, and lipids, crucial for cellular communication [105]. They are essential for regulating physiological processes like stem cell maintenance, tissue repair, and immune modulation, as well as contributing to disease pathogenesis [106]. Exosomes are highly promising for drug delivery due to their intrinsic material transport capabilities, long-term recycling potential, and favorable biocompatibility [107]. Exosomes, as a class of natural nanocarriers, demonstrate significantly superior biocompatibility and bioavailability compared to traditional synthetic nanocarriers produced in chemical environments [106]. Therefore, exosome-based therapy presents reduced concerns regarding immune rejection and tumorigenesis compared to stem cell therapy. Exosomes, due to their small size and cell-derived membranes, can be safely administered intravenously and possess natural targeting abilities [108]. Exosomes from bone marrow, adipose tissue, and umbilical-derived mesenchymal stromal cells (MSCs) have been studied for their potential in cardiac therapy [109]. Nevertheless, the complexity of the cargoes and low production yield are still unsolved problems which hinder the clinical translation of exosomes [110].

Figure 3E illustrates the nanotechnologies utilized for exosome isolation. The integration of microfluidic technology with nanotechnology has facilitated advancements in exosome enrichment [111]. Lee et al. pioneered the development of an acoustic nanofilter system capable of size-specific, continuous, and contact-free separation of exosomes. The system effectively isolated nanoscale vesicles from cell culture media and stored red blood cell products using exogenous acoustic waves, demonstrating the feasibility of exogenous-triggered isolations with high spatiotemporal precision [112]. Rho et al. created a magnetic nanosensor for detecting and profiling erythrocyte-derived exosomes [113]. As a notable example, Wunsch et al. designed nanoscale lateral displacement arrays capable of separating exosomes as small as 20 nm through displacement trajectories [114]. They employ silicon manufacturing techniques to create nanoscale deterministic lateral displacement arrays with consistent gap sizes between 25 and 235 nm. This configuration enables the high-resolution size-based isolation of exosomes within the 20 to 110 nm range [114].

### 2.6. Other Types

In addition to the aforementioned categories, there exist other forms of nanocarriers, including hybrid nanocarriers, cell membrane-coated nanocarriers, and high-density lipoprotein (HDL)-based nanocarriers, among others.

Hybrid nanocarriers integrate multiple nanoparticles into a functional nanoscale structure [115]. The main goal of research in this field is to create nanostructures that offer superior therapeutic efficacy beyond the simple combination of their individual components [115]. Lipid–polymer hybrid nanoparticles consist of a biodegradable polymeric core for encapsulating poorly water-soluble drugs, a lipid monolayer to enhance stability and reduce drug diffusion, and a lipid-PEG outer corona to protect against immune detection and prolong systemic circulation [116]. These nanoparticles exhibit robust mechanical integrity and in vivo stability, with enhanced drug encapsulation and an advantageous pharmacokinetic profile [117]. Furthermore, through the integration of mesoporous silica nanoparticles with various polymers (silicon–organic nanocarriers), research on controlled drug delivery systems has significantly advanced [117]. It has been reported that silica–polymer conjugates can modulate drug release, prevent premature leakage, sustain release, and enhance the therapeutic index while minimizing adverse effects [118,119].

Cell membrane-coated nanoparticles are created by covering synthetic cores with natural cellular membranes, forming a core–shell structure that imitates cellular characteristics [120,121]. These nanoparticles exhibit outstanding bio-interfacing abilities, allowing them to effectively navigate complex biological environments by avoiding immune clearance and selectively accumulating at disease sites [122]. Membranes from diverse cell types, such as red blood cells, platelets, macrophages, dendritic cells, neutrophils, natural killer cells, and T cells, have been used to fabricate these nanoparticles [123]. Notably, red blood cell membranes have been extensively utilized in applications where minimizing non-specific interactions is crucial, thereby serving as an effective alternative to synthetic PEG coatings [124]. Platelets, a type of anucleated blood cell, have been widely used as a source for nanoparticle surface coating membranes [123]. Given their pivotal role in hemostasis and their responsiveness to inflammatory signals, platelet membranes can be exploited for targeted delivery applications [125].

HDLs are natural nanoparticles, 7–13 nm in diameter, composed of diverse biological macromolecules [126]. HDLs are essential for reverse cholesterol transport, aiding in the removal of cholesterol from macrophage foam cells in atherosclerotic lesions [78]. HDLs are known to protect the endothelium by promoting nitric oxide production, exhibiting antioxidant properties that prevent LDL oxidation, and reducing inflammatory responses [127,128,129,130,131]. These attributes render HDL-based nanocarriers a promising strategy for targeting atherosclerosis [78].

**Figure 3 ijms-26-01743-f003:**
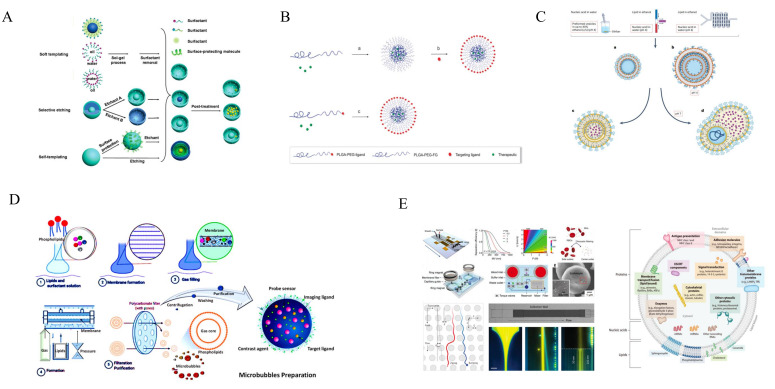
(**A**) Three novel synthesis approaches for hollow/rattle-type mesoporous silica nanoparticles [62]. (**B**) Self-assembly of polymer-based nanocarriers in aqueous solution [132]. (**C**) The synthesis processes of lipid-based nanocarriers [89]. (**D**) A diagrammatic representation of microbubble preparation and purification processes [99]. (**E**) The nanotechnologies utilized for exosome isolation [106].

## 3. Nanocarriers in Coronary Heart Disease

CHD is the leading cause of death and the main contributor to the global burden of disability-adjusted life years (DALYs). In 2015, CHD accounted for 8.9 million deaths and 164 million disability-adjusted life years (DALYs) [133]. CHD is characterized by the constriction or obstruction of the coronary arteries due to atherosclerosis, which consequently leads to diminished blood flow and oxygen deprivation in the myocardial tissue [134]. And this disease encompasses two critical pathological processes: atherosclerosis in its early stage and myocardial infarction in its late stage.

### 3.1. Nanocarriers in Atherosclerosis

In the early phase of atherosclerosis, atherogenic risk factors like smoking, hyperlipidemia, hypertension, diabetes, and aging may cause endothelial damage [135]. This damage enhances endothelial permeability, promoting the retention of low-density lipoprotein (LDL) particles. Activated endothelial cells secrete chemotactic chemokines and cytokines and express adhesion molecules, facilitating monocyte recruitment, attachment, and accumulation in the vascular wall. Monocytes differentiate into macrophages after ingesting apolipoprotein B-containing LDLs. This initial phase of lesion formation is marked by the subendothelial retention of lipoproteins and immune cells. Monocyte-derived macrophages ingest oxidized low-density lipoprotein (ox-LDL), becoming foam cells that accumulate in the vascular wall and promote atherosclerosis progression. Foam cell necrosis releases intracellular lipids into the lipid pool, forming the basis for atherosclerotic plaque development. In reaction to atherosclerotic stimuli, vascular media smooth muscle cells (SMCs) proliferate and migrate to the intima, forming fibrous caps. The fibrous cap, composed of SMCs, along with the lipid pool originating from foam cells, constitutes the primary component of atherosclerotic plaque [78,136]. Under normal conditions, atherosclerotic plaque remains stable when characterized by a thick fibrous cap and high collagen content. However, chronic inflammation and the presence of metalloproteinases can lead to collagen degradation and enlargement of the necrotic core, thereby transforming stable plaques into unstable ones, which may subsequently rupture. This progression may cause thrombosis and full coronary lumen blockage, leading to acute myocardial infarction onset [137]. The pathological processes of atherosclerosis and their associated gene therapy based on nanocarriers are summarized in Figure 4.

#### 3.1.1. Therapies Targeting Endothelial Cells in Atherosclerosis

As mentioned above, endothelial damage is pivotal in the early stage of the condition, and current pharmacological interventions primarily address systemic risk factors. Nevertheless, certain risk factors, such as age, are non-modifiable. Therefore, therapies specifically targeting endothelial cells are necessary. Previous research has identified several miRNAs that are implicated in critical cellular processes associated with atherosclerosis [138,139,140,141,142,143]. miR-92a is markedly upregulated in endothelial cells due to disturbed flow, which subsequently increases vascular inflammation [140,142,143,144]. Research indicates that administering naked miR-92a inhibitors to mice reduces atherosclerosis, suggesting its potential as a therapeutic target [142,143]. However, the systemic administration of miR-92a inhibitors has been shown to inhibit miR-92a across a broad spectrum of tissues beyond the vascular endothelium, which may lead to various adverse effects [143]. To solve this problem, Zhou et al. developed a tri-block polymer system (VHPKQHR-PEG-K30) for targeted delivery of miR-92a inhibitors to inflamed endothelium. Poly-L-lysine (K30) was conjugated to the terminal end of polyethylene glycol (PEG) to form a maleimide-terminated bio-macromolecular copolymer. A cysteine-modified VCAM-1-targeting peptide, VHPKQHR, was covalently linked to the PEG head using a thiol–maleimide reaction. In vivo experiments on atherogenic apolipoprotein E-deficient (ApoE−/−) mice showed that targeted micelles with miR-92a inhibitors reduced atherosclerotic lesions by 83% (*p* ≤ 0.001) compared to the PBS-treated control group and by 36% (*p* ≤ 0.01) compared to the group treated with the naked miR-92a inhibitor [21].

#### 3.1.2. Therapies Targeting Macrophages in Atherosclerosis

Leuschner et al. developed a liposomal nanoparticle encapsulating C-C chemokine receptor 2 (CCR2)-siRNA aimed at silencing CCR2 expression in monocytes. The nanoparticle improved siRNA delivery to monocytes, reducing their recruitment from the spleen and bone marrow, which resulted in a 38% decrease in atherosclerotic lesion size and a 34% reduction in myocardial infarct size in two distinct murine models (*p* < 0.05). Notably, this approach effectively targeted the recruitment mechanism while preserving the intrinsic immune function of immune cells [22]. In addition to diminishing the aggregation of monocyte-macrophages, the regulation of macrophage polarization and the inhibition of macrophage-mediated inflammation may represent other promising strategies. Teng et al. developed sialic acid-coated baicalein nanorods (BNRs) encapsulating antisense oligonucleotides targeting miR-155, with baicalein functioning as both a delivery vehicle and anti-inflammatory agent. Overexpressing miR-155, a key microRNA, results in the reduced activity of mechanistic target of rapamycin complex 1 (mTORC1) kinase and Ras homolog enriched in the brain. This overexpression enhances M1 macrophage polarization, worsening inflammation and disease progression. The anti-miR155-targeting nanomedicine significantly decreased miR155 expression and M1 macrophage prevalence, reducing inflammation and enlarging artery lumen diameter compared to the control group (*p* < 0.001) [23]. Moreover, Bu et al. developed an IL-10 mRNA-encapsulating exosome, inspired by the hepatitis C virus internal ribosome entry site. miR-155 activates IL-10 mRNA in vivo, resulting in Interleukin-10 (IL-10) protein synthesis. miR-155, influenced by in vivo inflammation levels, triggers IL-10 mRNA translation. This nanoparticle enables IL-10 delivery contingent on miR-155 levels, providing a responsive delivery system. This nano-delivery approach successfully slowed plaque progression in ApoE−/− mice and reduced the usual side effects of systemic IL-10 administration. The expression levels of pro-inflammatory cytokines IL-1β, IL-6, and TNF-α, along with atherosclerotic lesion size, were significantly reduced (all *p* < 0.05), as demonstrated by Oil Red O staining [24].

To prevent plaque necrosis, a potential strategy could involve augmenting macrophage efferocytosis to facilitate the clearance of necrotic cells. A prior study demonstrated that activating the liver X receptor-alpha (LXRα)/ATF6/c-MerTK signaling pathway enhances efferocytosis and the removal of necrotic cells from plaques. The Ca2+/calmodulin-dependent protein kinase γ (CaMKIIγ) inhibits the MerTK pathway activation, leading to secondary necrosis of untreated plaque necrotic cells. This process ultimately contributes to plaque necrosis and the thinning of fibrous caps, rendering the plaque unstable [145]. This instability may cause plaque rupture, acute intravascular thrombosis, and other acute arteriosclerotic events [146]. Thus, reducing CaMKIIγ expression in damaged macrophages offers promising therapeutic potential for treating advanced arteriosclerosis. The gene Camk2g, which encodes CaMKIIγ in macrophages, was identified as a target for intervention. Camk2g-siRNA was encapsulated in a lipid-based nanocarrier conjugated with the peptide ligand S2p for targeted macrophage delivery by Tao et al., and PEG was added to prolong the circulation time in the bloodstream. In vivo studies on Ldlr-deficient mice on an eight-week Western diet showed that S2P-siCamk2g nanoparticles significantly improved efferocytosis, decreased necrotic core areas (*p* < 0.05), and increased collagen cap thickness (*p* < 0.05) by modulating CaMKIIγ and MerTK expression in macrophages [25].

#### 3.1.3. Therapies Targeting Formation of Foam Cells in Atherosclerosis

The internalization of oxidized low-density lipoprotein (ox-LDL) is crucial for the transformation of macrophages into foam cells. The membrane receptors Cluster of Differentiation (CD36) and SR-A1 are pivotal in facilitating the internalization of ox-LDL. CD36, part of the scavenger receptor class B (SR-B) family, facilitates the internalization of oxidized low-density lipoprotein (ox-LDL) by forming CD36-oxLDL complexes. SR-A1 acts as a lipid transport receptor regulated by the nuclear factor kappa B (NF-κB ) signaling pathway and is involved in the internalization of ox-LDL [147,148]. In atherosclerosis models using ApoE−/− mice, the knockdown of SR-A1 has been shown to inhibit ox-LDL internalization and the subsequent formation of foam cells [149]. Consequently, CD36 and SR-A1 represent critical therapeutic targets for the development of nanomaterials in preclinical studies aimed at inhibiting lipid internalization. According to these findings, Jiang et al. used engineered biomimetic nanomaterials for targeted delivery of SR-A siRNA and pitavastatin, featuring phosphatidylserine surface modification to selectively target diseased tissues. A three-month treatment regimen resulted in a significant 65.8% reduction in plaque area relative to the control group (*p* < 0.01) [26]. Smooth muscle cell metaplasia can result in the formation of foam cells akin to macrophages [150]. miRNA-145 (miR-145) is integral to the phenotypic transformation of smooth muscle cells [151], highlighting its potential as a therapeutic target. Chin et al. developed a nano-delivery system, 1,2-distearoyl-sn-glycero-3-phosphoethanolamine-N-[(polyethylene glycol)-2000] (DSPE-PEG (2000))-miR-145, for in vivo miR-145 delivery. The nanoparticles were modified with monocyte chemoattractant protein-1 (MCP-1) peptide to target C–C chemokine receptor-2 (CCR2), abundantly present on synthetic vascular SMCs. According to the in vivo experiments conducted on ApoE−/− mice, treatment with miR-145 nanoparticles resulted in a 49 ± 0.1% reduction in lesion size (*p* < 0.01) and a 73 ± 0.3% reduction in the necrotic core area (*p* < 0.001) of the ascending aorta compared to those treated with PBS [27].

#### 3.1.4. Therapies Targeting Cholesterol Metabolism in Atherosclerosis

Reverse cholesterol transport (RCT) is an essential physiological mechanism that facilitates the movement of cholesterol from the bloodstream to the liver for metabolism and excretion. High-density lipoprotein (HDL) facilitates this process, mediated by cholesterol transport proteins including ATP-binding cassette A1 (ABCA1), ATP-binding cassette transporter G2 (ABCG2), and SR-B1. Liver X receptors (LXRs) are crucial transcription factors in lipid metabolism, responsible for activating ABCA1 expression. ABCA1, linked to HDL, is essential for transferring cholesterol to apolipoprotein A-1 (ApoA-1) during reverse cholesterol transport (RCT), aiding in the clearance of cholesterol from atherosclerotic plaques. Due to its significant role in RCT, ABCA1 is widely recognized as a key target in atherosclerosis therapy [30]. Several miRNAs, including miR-206, miR-223, and miR-33a, regulate cholesterol efflux by directly interacting with ABCA1. Nguyen et al. developed a technique to encapsulate miR-206 and miR-233 in chitosan nanoparticles. Chitosan nanoparticles improved miRNA loading capacity and stability, thereby enhancing RCT efficacy [28]. A nano-drug delivery system was advanced by integrating antagomiR-33a therapeutic molecules with recombinant high-density lipoprotein (r-HDL) as the carrier [29]. As presented by JB et al., r-HDL nanoparticles demonstrate excellent biocompatibility and a targeted affinity for the HDL receptor, enabling the intracellular delivery of antagomiR-33a, which enhances ABCA1 gene expression to promote RCT. Meanwhile, r-HDL is capable of directly sequestering cholesterol, thereby promoting its transport and excretion [29]. Interestingly, He et al. employed an approach that integrated RCT with the inhibition of ox-LDL internalization. The study involved engineering dendritic nanoparticles to deliver LXR ligands and SR-A-siRNA. Activation of LXRs was intended to modulate ABCA1/ABCG1 to facilitate RCT, while SR-A-siRNA was aimed to downregulate SR-A receptor expression, thereby reducing ox-LDL internalization. These nanoparticles were further functionalized with mannose to specifically target plaque tissue, which is characterized by high mannose receptor expression. Through this combination therapy, a reduction in the total (29.9 ± 7.1 vs. 17.4 ± 3.7, *p* < 0.0001) and aortic arch (54.6 ± 8.6 vs. 36.2 ± 8.1, *p* < 0.0001) plaque area was observed compared to the untreated group [30].

PCSK9, a liver-secreted protease, plays a key role in cholesterol metabolism by binding to LDL receptors (LDL-Rs) both outside and inside cells to promote their degradation. Inhibition of this protease prevents LDL-R degradation, promoting their recycling and significantly improving LDL-C clearance [152]. Recent years have seen substantial advancements in gene therapy targeting PCSK9. Although animal models and an early-stage clinical trial showed satisfactory safety and efficacy for intravenous siRNA-PCSK9 encapsulated in lipid-based nanocarriers (ALN-PCS) [31,153], the clinical trial was discontinued after Phase I. The discontinuation was mainly attributed to the introduction of PCSK9-directed monoclonal antibodies, which are administered subcutaneously every 2 to 4 weeks, providing a more advantageous administration method and making ALN-PCS commercially unfeasible [20]. However, with the recent advancements in the N-acetyl-galactosamine (GalNAc) platform, siRNA-PCSK9-based therapy has once again garnered the attention of researchers. The initial trial results for inclisiran, a small interfering RNA which inhibits PCSK9 synthesis and is delivered by the GalNAc system, were published in 2017 [154]. The study demonstrated that a single 300 mg dose or higher of inclisiran can decrease circulating PCSK9 levels by around 75% and reduce LDL-C levels by approximately 51%, with these effects lasting over 6 months after injection. Adverse events were mostly mild to moderate, with no serious incidents or discontinuations due to adverse effects [154]. At present, inclisiran has been approved in the European Union, the United States, and several other countries, based on efficacy and safety data from earlier trials [20].

VERVE-101 is an experimental CRISPR base editing therapy aimed at inducing a specific adenine-to-guanine nucleotide change at a splice donor site in the PCSK9 gene within liver cells, permanently deactivating its expression and leading to a long-term decrease in LDL cholesterol levels. This treatment includes mRNA for an adenine base editor protein and guide RNA targeting the PCSK9 gene, both delivered in an LNP through intravenous infusion [20]. In a study conducted with 36 cynomolgus monkeys, a single intravenous dose of VERVE-101 at 1.5 mg/kg led to an average reduction of 83% in blood PCSK9 protein levels and a 69% decrease in LDL cholesterol, with these effects persisting for up to 476 days post-administration. Liver safety assessments showed only temporary increases in alanine aminotransferase (ALT) and aspartate aminotransferase (AST) levels in both the control and VERVE-101-treated monkeys, with no changes in total bilirubin levels, suggesting no effect on overall liver function [32]. VERVE-101 is currently undergoing a clinical trial (NCT05398029) for patients with heterozygous familial hypercholesterolemia, established atherosclerotic cardiovascular disease, and uncontrolled hypercholesterolemia, despite maximum tolerated oral lipid-lowering therapy. This trial represents the first in vivo use of base editing technology in humans. Recently, the interim results of this trial indicated that a total of ten patients were treated across four dose cohorts, with a mean age of 54 years and a baseline LDL-C of 193 mg/dL. Participants received a single peripheral intravenous infusion of VERVE-101, preceded by antihistamines and dexamethasone (Dex). The treatment led to a sustained significant reduction in PCSK9 levels by 47%, 59%, and 84% and in LDL-C levels by 39%, 48%, and 55% among three participants in the high-dose cohorts (0.45 and 0.6 mg/kg). All three patients encountered infusion site reactions and temporary ALT elevations. Additionally, cardiovascular adverse events were observed in two participants [155]. Future steps involve increasing cohort enrollment for high-dose groups and conducting a Phase II randomized placebo-controlled clinical trial.

### 3.2. Nanocarriers in MI

Myocardial infarction (MI), the most severe form of CHD, is a major life-threatening condition often linked to sudden cardiac death [156]. The gold-standard treatment for myocardial infarction, supported by adjunctive therapies, emphasizes rapid blood flow restoration via percutaneous coronary intervention (PCI) or thrombolytic agents, potentially minimizing myocardial infarct size and enhancing clinical outcomes [157]. Restoring coronary blood flow may lead to ischemia/reperfusion (I/R) injury. Myocardial I/R injury is characterized by the release of inflammatory cytokines and chemokines that recruit circulating leukocytes, mainly neutrophils and monocytes, to the ischemic area. These recruited cells subsequently infiltrate the myocardial tissues. Myocardial I/R injury is notably characterized by significant inflammation. Beyond inflammation, I/R injury triggers excessive reactive oxygen species (ROS) production and calcium ion (Ca2+) overload, activating multiple cell death pathways, including apoptosis, autophagy, ferroptosis, necroptosis, and other regulated mechanisms. The loss of cardiomyocytes finally leads to impairment of cardiac function and irreversible ventricular remodeling [108,158].

#### 3.2.1. Therapies Targeting Inflammation in Myocardial Infarction

Hou et al. synthesized cRGD-modified, PEGylated, ditellurium-crosslinked polyethylenimine (RPPT) by crosslinking PEI 600 with ditellurium and then modifying it with PEG and the endothelial cell-targeting peptide cyclic arginine-glycine-aspartic acid (cRGD). Subsequently, RPPT was employed to encapsulate PLGA nanoparticles loaded with Dex and to condense VCAM-1 siRNA. In rats with myocardial I/R injury, systemically administered cRGD-modified nanocarriers successfully targeted and penetrated inflamed endothelial cells. Within these cells, RPPT was degraded by high levels of reactive oxygen species (ROS), promoting the intracellular release of VCAM-1 siRNA and improving VCAM-1 silencing efficiency. Due to the synergistic effects of DXM and VCAM-1 siRNA, the nanoparticles significantly reduced neutrophil infiltration into the ischemic myocardium, thereby demonstrating strong anti-inflammatory efficacy in mitigating myocardial I/R injury and promoting the recovery of cardiac function. The treatment with nanoparticles significantly reduced infarct size (8.1% vs. 45.1%, *p* < 0.01), fibrosis area by 13.4%, and apoptotic cell percentage by 20.8% (both *p* < 0.001) [37]. Similarly, the regulation of macrophage polarization could also be another promising strategy in MI. miRNA-21 mimics were encapsulated in nanoparticles formed by the spontaneous complexation of hyaluronan-sulfate (HAS) with the nucleic acid, mediated by calcium ion bridges. Following intravenous administration, the miRNA-21 nanoparticles selectively targeted post-myocardial infarction macrophages within the infarct region. These nanoparticles prompted macrophages to transition from a pro-inflammatory to a reparative phenotype, which improved angiogenesis and reduced hypertrophy, fibrosis, and apoptosis in the remote myocardium, as demonstrated by Bejerano et al. [36]. Shen et al. investigated the influence of MSC-derived exosomes on M2 macrophage differentiation through miR-21-5p, alongside a reduction in inflammatory factor expression [35]. In a study presented by Zhao et al., in murine models, intra-myocardial administration of mesenchymal stem cell-derived exosomes (MSC-exo) effectively polarized macrophages to the M2 phenotype. This polarization contributed to the attenuation of the inflammatory response and facilitated myocardial repair. Seven days after myocardial infarction, both serum and heart tissues showed significantly decreased IL-6 and increased IL-10 concentrations compared to the control group (all *p* < 0.0001). Additionally, the infarct size was reduced as well (*p* < 0.0001). Further analyses and validation experiments revealed that miRNA-182 (miR-182) is pivotal in the polarization process by regulating the TLR4/NF-κB/PI3K/Akt signaling pathway [34]. Additionally, exosomes from hypoxic bone marrow-derived mesenchymal stem cells (BMSCs) containing miR-98-5p have been shown to reduce myocardial enzyme levels, oxidative stress, inflammatory responses, macrophage infiltration, and infarct size in ischemia/reperfusion (I/R) myocardial tissue. This effect occurs by inhibiting toll-like receptor 4 (TLR4) and activating the PI3K/Akt signaling pathways in rat models of MI/RI [33].

#### 3.2.2. Therapies Targeting Cardiomyocyte Apoptosis and Myocardial Repair After Myocardial Infarction

miRNAs play a significant role in angiogenesis, apoptosis, fibrosis, and arrhythmogenesis following myocardial infarction [159,160]. Exosomes are natural nanocarriers for miRNAs in vivo. The study presented by Wang et al. evaluated the therapeutic effects of mesenchymal stem cells (MSCs) from endometrium, bone marrow, and adipose tissue on infarcted rat myocardium, finding that exosomes from endometrial MSCs demonstrated superior efficacy. The improved effectiveness is probably due to the paracrine-induced increase in miR-21 expression, which is essential for enhancing cell survival and angiogenesis [161]. Studies using modified oligonucleotides (agomir) in murine I/R myocardium models, alongside MSCs or MSC-derived exosomes, have shown that miR-125a-5p agomir influences macrophage, fibroblast, and cardiomyocyte functions. This therapeutic approach improves cardiac function and remodeling in porcine myocardial I/R models without raising arrhythmia rates or systemic toxicity [44]. Exosome composition and biological activity are largely determined by their originating donor cells. Interestingly, Sun et al. isolated exosomes from young (Young-Exo) and aged (Age-Exo) MSCs to evaluate their regenerative potential [43]. The study revealed that Young-Exo MSCs were more effective than Age-Exo MSCs in enhancing endothelial tube formation, decreasing fibrosis, and preventing cardiomyocyte apoptosis in vitro. In vivo results demonstrated a significant improvement in left ventricular ejection fraction (LVEF) and left ventricular fractional shortening (LVFS) in the Young-Exo group compared to the MI group at 2 weeks (*p* < 0.05) and 4 weeks (*p* < 0.01) post-MI. The Young-Exo group exhibited a significant reduction in fibrosis and collagen areas (*p* < 0.01 and *p* < 0.001, respectively). The findings demonstrated that Young-Exo MSCs enhanced cardiac structure and function in rat hearts post-MI. Further analysis revealed that miR-221-3p plays a crucial role in enhancing Akt kinase activity by inhibiting phosphatase and tensin homolog (PTEN) [43].

Besides exosomes, various nanocarriers have also been utilized for miRNA delivery in MI. miRNA-133 (miR-133) inhibits myocardial cell apoptosis, reduces myocardial infarction, and provides cardioprotection, highlighting its potential as a therapeutic target for MI [162,163,164,165]. Sun et al. utilized arginine-glycine-aspartic acid (RGD)-modified PEG-PLA as a delivery system for miR-133 to target cardiac lesions. Following nanoparticle treatment, the left ventricular end-systolic diameter (LVESD), left ventricular end-diastolic diameter (LVEDD), and left ventricular posterior wall thickness at end-systole (LVPWS) were significantly reduced (all *p* < 0.01), while left ventricular ejection fraction (LVEF) and left ventricular fractional shortening (LVFS) were significantly increased (both *p* < 0.01) compared to the model group. The findings suggest that RGD-PEG-PLA/miR-133 alleviated myocardial damage and reduced cell apoptosis in MI rats, helping to maintain left ventricular morphology and function [42]. miRNA-21 (miR-21) shows promise in reducing myocardial infarct size and cardiomyocyte apoptosis during acute myocardial infarction [166], and aids cardiovascular regeneration by regulating target genes like programmed cell death 4 (PDCD4) [167,168]. Liposomes were employed by Li et al. to encapsulate miR-21, with their surfaces modified using an anti-cardiac troponin T (cTnT) antibody to specifically target ischemic myocardial tissues (these nanoparticles are termed cT-21-LIPs). Intravenous cT-21-LIPs significantly improved LVEF and LVFS at 3 and 15 days post-MI (both *p* < 0.001) and reduced infarction size compared to the MI group (16.4% vs. 38.3%, *p* < 0.001) [41]. miRNA-1 (miR-1) is a potential target for anti-arrhythmic therapy, as its repression can alleviate arrhythmia [160]. Liu et al. explored the targeted delivery of oligonucleotides to ischemic myocardial tissues using liposomes modified with anti-cardiac troponin I antibodies and loaded with anti-miR-1 antisense oligonucleotides (AMO-1) (these nanoparticles are termed cT-A-LIP). The study showed that cT-A-LIP effectively alleviated ischemic arrhythmia by silencing miR-1 in ischemic myocardium and restoring the depolarized resting membrane potential (RMP) in myocardial infarction (MI) rats [40].

The receptor for advanced glycation end products (RAGE), a pattern recognition receptor, is essential in regulating pro-inflammatory and pro-apoptotic processes in I/R-injured myocardium by accumulating ligands at inflammation sites [169,170,171]. In a study presented by Lan et al., cardiomyocyte-specific MSNs were designed for the ROS-responsive co-delivery of Dex and siRNA-RAGE to alleviate myocardial inflammation [39]. The nanosystem’s ability to respond to reactive oxygen species (ROS) is due to the surface-decorated PPTP, an ROS-degradable polycation derived from PGE2-modified, PEGylated, and ditellurium-crosslinked polyethylenimine (PEI). The MSN drug delivery system enabled the concurrent release of Dex and siRNA-RAGE in myocardial areas with high ROS levels, achieving substantial RAGE silencing (72%) and improved anti-inflammatory outcomes. Consequently, these nanoparticles substantially reduced myocardial fibrosis and apoptosis, ultimately leading to the restoration of systolic function [39]. In addition, Yu et al. employed UTMD with the phSDF-1α-NF-κB plasmid to enhance the transfection efficiency of the SDF-1α gene. The results demonstrated that the upregulation of the SDF-1α gene significantly increased angiogenesis and myocardial perfusion as well as concentrations of norepinephrine and N-terminal pro-B-type natriuretic peptide (NT-proBNP) from one day to one month after MI, preserving the left ventricle from remodeling and consequently improving cardiac function. Immunostaining and Masson’s trichrome staining revealed a significant increase in microvascular density (46.4 ± 6.8 vs. 4.1 ± 1.8, *p* < 0.001) and a significant decrease in infarct size (46.5% ± 1.3% vs. 53.7% ± 2.1%, *p* < 0.01) in peri-infarct regions after nanoparticle treatment [38].

## 4. Nanocarriers in Pulmonary Hypertension

Pulmonary hypertension (PH) is defined by a mean pulmonary artery pressure of 25 mm Hg or greater at rest, as determined through right heart catheterization [172]. Affecting approximately 1% of the global population, it is a significant health issue impacting all age groups [173]. Pulmonary hypertension is classified into five groups based on pathophysiological, clinical, and therapeutic characteristics: pulmonary arterial hypertension, pulmonary hypertension due to left-sided heart disease, pulmonary hypertension due to lung disease or hypoxia, chronic thromboembolic pulmonary hypertension, and pulmonary hypertension with unclear or multifactorial mechanisms [173]. This review mainly focuses on the group of pulmonary arterial hypertension (PAH).

Pulmonary arterial hypertension is a rare, complex, severe, life-threatening, and progressive disease characterized by marked pulmonary vascular remodeling [174,175]. Pulmonary vascular endothelial dysfunction is crucial in the onset and progression of PAH, characterized by pulmonary arterial endothelial cells (PAECs) exhibiting a pro-inflammatory and apoptosis-resistant proliferative phenotype, alongside undergoing endothelial-to-mesenchymal transition (EndoMT) [176]. The expression of pro-inflammatory cytokines (IL-1, IL-6, IL-8, IL-12, and C-C motif chemokine ligand 2 (CCL2)) and proteins related to leukocyte recruitment and adhesion (E-selectin, leptin, macrophage migration inhibitory factor, intercellular adhesion molecule-1, and vascular adhesion molecule-1) is increased in these abnormal PAECs [176,177,178,179]. Inflammatory cell infiltration is a pathological feature of remodeled pulmonary vasculature, with inflammation playing a crucial role as evidenced by the correlation between circulating inflammatory markers and disease severity [180], the presence of organized lymphoid follicles near pulmonary arterial hypertension lesions [181], and the frequent association of PAH with autoimmune or inflammatory diseases [182]. Targeting inflammation caused by endothelial dysfunction and partially driven by NF-κB activation [183] may be essential for treatment. EndoMT represents a pathological alteration in PAECs linked to the dysregulation of the transforming growth factor-beta (TGF-β)/bone morphogenetic protein (BMP) signaling pathway [184]. Abnormal PAECs result in excessive production of vasoconstrictors like endothelin (ET) and/or decreased synthesis of vasodilators such as nitric oxide (NO) or prostaglandin I2 (PGI2) [185]. Current pharmacological treatments, including prostacyclin analogs or receptor agonists, phosphodiesterase-5 inhibitors, guanylate cyclase activators, endothelin receptor antagonists, and calcium channel blockers, aim to target three interrelated signaling pathways [186].

Dynamic and maladaptive remodeling of the pulmonary vasculature involves the emergence and maintenance of a pro-proliferative and apoptosis-resistant phenotype in various contractile vascular cells within the arterial wall, such as pulmonary artery smooth muscle cells (PASMCs), adventitial fibroblasts, and cells with myofibroblast-like characteristics, in addition to pulmonary artery endothelial cells [187,188,189]. In adult physiological conditions, most PASMCs are either quiescent or maintain a contractile function, representing their differentiated phenotype [190,191]. These cells are morphologically spindle-shaped or elongated with blunt-ended or cigar-shaped nuclei. The shift of PASMCs from a contractile to a synthetic phenotype is mainly driven by growth factors, inflammatory mediators, mechanical stimuli changes, epigenetic modifications, and ECM composition and organization alterations [192]. During this transition, PASMCs develop enhanced migratory and proliferative abilities, alongside resistance to apoptosis. The pathological process involves intrinsic abnormalities such as bone morphogenic protein type II receptor (BMPRII) signaling dysfunction or loss, potassium channel homeostasis dysregulation, reduced peroxisome proliferator-activated receptor gamma (PPAR-γ) and Forkhead box protein O1 (FOXO1) pathway signaling, constitutive NF-κB and hypoxia-inducible factor-1 alpha (HIF-1α) activation, and ion channel remodeling. This is evidenced by disrupted calcium and potassium homeostasis, altered calcium sensitivity, and transcription factor activation [187,192]. The phenotypic transition of PASMCs is linked to metabolic reprogramming, marked by a shift from mitochondrial oxidative phosphorylation to glycolysis, and disruptions in amino acid and lipid metabolism [193].

Despite extensive efforts over the years, the currently available pharmacological treatments, primarily vasodilators as previously noted, do not effectively inhibit the vascular remodeling process. The mortality rate is still high, though the median survival of patients with PAH has improved from 2.8 to approximately 7 years due to advancements in treatment, with outcomes varying based on etiology, treatment response, and patient-specific factors [5,6,7,8]. Consequently, it is crucial to create new PAH treatments based on cutting-edge technology and a deeper comprehension of the disease’s pathogenesis. Gene therapies based on nanocarriers and their therapeutic targets in PAH are summarized in Figure 5.

### 4.1. Therapies Targeting Pulmonary Artery Endothelial Cells

Pulmonary artery endothelial cells (PAECs) play a key role in the onset and development of pulmonary arterial hypertension (PAH), making them a focal point of research. The NF-κB and BMP signaling pathways have been identified as major therapeutic targets. In a study presented by Satoshi Kimura et al., NF-κB “decoy” oligodeoxynucleotides were loaded on bioabsorbable polymeric nanoparticles which were formulated from a poly-(ethylene glycol)-block-lactide/glycolide copolymer (PEG-PLGA). The study demonstrated that intratracheal instillation of polymeric nanoparticles mitigated inflammation, reduced small pulmonary arterial remodeling, and decreased right ventricular systolic pressure (RVSP), an indirect measure of pulmonary arterial systolic pressure [52]. CLIC4 (Chloride Intracellular Channel 4), part of the CLIC protein family (CLIC1-6), is primarily found in the pulmonary vascular endothelium and is essential for controlling cell proliferation, apoptosis, angiogenesis, and differentiation [194,195]. A prior study has demonstrated that the expression of CLIC4 was upregulated in the lungs of both PAH patients and PAH animal models [196]. Composed of a cationic lipid, cholesterol, and 1,2-distearoyl-sn-glycero-3-phosphoethanolamine-N (methoxy (polyethylene glycol)-2000), DACC is a novel cationic lipoplex nanoparticle delivery system developed by V Fehring et al., which demonstrates great pharmacokinetic properties and effective cellular uptake by pulmonary endothelial cells [197]. This nanocarrier was further used by Vahitha B et al., and the results showed that in vivo administration of CLIC4 siRNA significantly enhanced the expression of BMP type 2 receptor, diminished the activation of NF-κB in the pulmonary arterial endothelium, and consequently reduced RVSP [51]. While the role of bone morphogenetic protein signaling via the BMP type 2 receptor in the EndoMT of PAECs has been extensively studied over time, the significance of BMP type 1 receptors in this process has only recently been investigated. Heon-Woo Lee et al. demonstrated that the loss of endothelial BMP type receptor 1A (BMPR1A) triggers EndoMT by upregulating transforming growth factor-beta type II receptor (TGFBR2) and activating TGF-β signaling pathways. Intravenous administration of TGFBR2 siRNA encapsulated in 7C1 nanocarriers significantly reduced PAEC muscularization and decreased RVSP in the endothelial deletion of (Bmpr1a^iECKO^) mice [50]. The 7C1 nanocarrier, targeting PAECs, was synthesized by reacting C15 epoxide-terminated lipids with PEI600 at a 14:1 molar ratio and formulated with C14PEG2000 [198]. In addition, exosomal miRNAs miR-181a-5p and miR-324-5p, induced by Krüppel-like factor 2 (KLF2), play a crucial role in maintaining pulmonary vascular endothelium homeostasis, partially mediated by Notch4 and ETS-1. The intravenous delivery of miR-181 and miR-324 within a lipid-based carrier significantly decreased RVSP and pulmonary vascular muscularization in an animal model [49].

### 4.2. Treatments Aimed at Pulmonary Artery Smooth Muscle Cells

In addition, the proliferation of PASMCs has been another therapeutic target in several studies. In a study presented by Chao Teng et al. [48], baicalein, a small molecule with anti-inflammatory properties, was utilized to synthesize rod-like crystals to serve as carriers to co-deliver the pDNA of p53, which is a tumor suppressor gene capable of inducing apoptosis of cells [199] but found to be defective in PAH PASMCs [200]. And these nanoparticles were further coated by glucuronic acid via ionic complexation for targeting glucose transport-1 (GLUT-1) on PASMCs. The study demonstrated that administering these nanoparticles inhibited excessive PASMC proliferation in monocrotaline (MCT)-induced rats by activating the Bax/Bcl-2/Cas-3 apoptosis pathway, leading to a significant reduction in mean pulmonary artery pressure (mPAP) [48]. The TRPC gene, encoding a light-activated ion channel in photoreceptors, is a potential target for PASMCs. Mammalian TRPC homologs constitute a superfamily of six cation channel subtypes, with TRPC1 and TRPC6 crucially involved in modulating smooth muscle cell contraction and proliferation in murine models of hypoxia-induced PAH, as shown in prior research [201,202]. In hypoxia-treated animals, Cheuk-Kwan Sun et al. showed that the intratracheal administration of siRNA-TRPC1 using Lipofectamine effectively inhibited TPRC1 protein expression and significantly decreased RVSP in the lung [47]. Additionally, miRNAs like miR-145 and miR-204 have been identified as contributors to PASMC proliferation. In both PAH patients and animal models, miR-145 expression is upregulated, playing a crucial role in the muscularization of pulmonary arteries in mice under chronic hypoxia conditions [203]. Jared M. McLendon et al. employed Star:Star-mPEG-550, a functionalized cationic lipopolyamine, to target pulmonary vascular cells for delivering an antisense oligonucleotide against miR-145. Administration of the antisense oligonucleotide against miR-145 reduced pulmonary artery wall thickness and occlusive vascular lesion density without evident toxicity in other organs [46]. Furthermore, research indicated that miR-204 downregulation correlates with PAH severity and promotes proliferative and anti-apoptotic characteristics in PASMCs. And the pulmonary arterial pressure was reduced by intratracheal nebulization of synthetic miR-204 loaded on lipid-based carriers in MCT-induced PAH rats [45]. Notably, miR-145 inhibitor and synthetic miR-204 exhibited efficacy in established lesions, suggesting these two miRNAs might be potential therapeutic targets in the intermediate and even the late stage of PAH.

## 5. Nanocarriers in Hypertension

Hypertension, known as “the silent killer” for its early asymptomatic nature, significantly increases the risk of coronary artery disease, heart failure, cerebral hemorrhage, and chronic renal failure if not properly managed [204]. Around 1.4 billion people worldwide suffer from hypertension, with thiazide diuretics, ACE inhibitors, and calcium channel blockers being the most commonly recommended first-line treatments. Nevertheless, only a small proportion (approximately 14%) of patients achieve effective blood pressure control [204].

In recent years, the perspective that hypertension constitutes a syndrome rather than a singular disease, characterized by the common manifestation of elevated blood pressure, has gained wider acceptance. This suggests that the pathophysiological mechanisms underlying hypertension may vary significantly among individuals, and the absence of therapies targeting specific etiological factors largely contributes to the suboptimal treatment success rates. Therefore, developing more precise therapeutic strategies for hypertension is urgently needed. Worapaka Manosroi et al. categorized primary hypertension into four tiered subgroups in a recent review: (a) distant phenotype (hypertension), (b) intermediate phenotype (salt-sensitive and salt-resistant hypertension), (c) subintermediate phenotypes within salt-sensitive hypertension (normal renin and low renin), and (d) proximate phenotypes (genotype-specific hypertensive subgroup). After selection with strict criteria, a total of 21 genes associated with hypertension were identified. Eighteen genes were associated with salt-sensitive hypertension, with four linked to normal renin levels, eight to reduced renin levels, and six to indeterminate renin levels. The ACE gene, a well-researched gene associated with hypertension, encodes an enzyme that converts angiotensin I into angiotensin II. The gene was identified in a subgroup with an undefined renin, salt-sensitive, subintermediate phenotype of primary hypertension. The angiotensin receptor type I gene (AGTR1), responsible for encoding angiotensin receptor type 1 (AT1R), was found in the subgroup with an unspecified hypertension phenotype [4].

With advancements in nanotechnology, gene therapy has emerged as a promising therapeutic approach for hypertension. In a study conducted by M. N. Repkova et al., an ACE gene-targeted antisense oligonucleotide was conjugated with polylysine and immobilized on titanium dioxide nanoparticles. Nanoparticles containing antisense oligonucleotides were administered to the rat model of inherited stress-induced arterial hypertension (ISIAH), using either intraperitoneal injection or inhalation. Both administration routes significantly reduced systolic blood pressure by 20–30 mmHg compared to control groups, with variations potentially due to differences in dosage or pharmacokinetic properties [53]. A related study employed silicon–organic nanocarriers in place of titanium dioxide due to their ability to form stable solutions and reduced tendency to aggregate during storage. Antisense oligodeoxyribonucleotides targeting angiotensin-converting enzyme and angiotensin receptor type 1 mRNA were electrostatically immobilized on silicon–organic nanocarriers and administered to hypertensive ISIAH rats. This method led to a notable decrease in systolic blood pressure by around 30 mmHg [54].

Notably, the application of nanocarrier-based gene therapy resulted in a significant and sustained reduction in blood pressure compared to conventional treatments, which require daily administration. Furthermore, this therapeutic approach demonstrated high specificity and mitigated numerous adverse effects commonly associated with traditional pharmaceuticals. Although the precise gene therapy for hypertension is still at the initial stage and needs more investigations, its future is promising as the emerging nanocarriers do provide much facilitation.

## 6. Nanocarriers in Valvular Heart Diseases

Valvular heart disease arises from the dysfunction of one or more cardiac valves. This prevalent cardiovascular disorder affects approximately 5–10% of individuals aged 65–74 years and 10–20% of those over 75 years, with its incidence escalating rapidly, especially in high-income nations. Calcific aortic valve disease (CAVD) is a prevalent valvular heart condition in the elderly, affecting approximately 9.4 million individuals worldwide in 2019 [205]. CAVD encompasses a range of pathological conditions, beginning with aortic valve thickening and calcification commonly termed aortic sclerosis, and extending to aortic stenosis [205]. Both moderate and severe forms of aortic stenosis are linked to elevated morbidity and mortality rates, with five-year mortality rates of 56% and 67%, respectively [206]. Nonetheless, no medical therapies have been developed to decelerate disease progression, and the prognosis remains poor for individuals considered unsuitable for surgical or transcatheter interventions, with a 5-year mortality rate of 94% [207].

Significant advancements have been achieved in elucidating the pathophysiology of this disease over recent decades. Aortic valves are primarily composed of valvular endothelial cells (VECs) and valvular interstitial cells (VICs). The EndoMT of VECs, along with the myofibroblastic and osteogenic differentiation of VICs, is recognized as a pivotal mechanism underlying CAVD. The pathological process of CAVD can be divided into two distinct phases, which are described in detail below. Endothelial dysfunction, triggered by mechanical and oscillatory shear stress or other risk factors, promotes lipid deposition and immune cell infiltration, initiating CAVD pathogenesis. Elevated oxidative stress oxidizes depositional lipids into ox-LDLs and oxidized phospholipids, triggering chronic inflammation by upregulating cell adhesion molecules (e.g., ICAM-1, VCAM-1) and activating local immune cells (e.g., macrophages, CD4+ and CD8+ T lymphocytes). Following the initial inflammatory phase characterized by lipid deposition and chronic inflammation, the disease progresses to a stage where calcification becomes the predominant factor. During the propagation phase of CAVD, the differentiation of VICs, including those originating from VECs via EndoMT, into myofibroblast-like and osteoblast-like phenotypes is a critical process. Several signaling pathways have been implicated in this stage. The TGF-β1/small mothers against decapentaplegic (Smad)3 pathway is associated with the EndoMT of VECs and myofibroblastic differentiation of VICs, whereas the WNT3A/β-catenin pathway promotes Runx2 expression, a crucial transcription factor in osteogenesis and ectopic mineralization, thus facilitating VICs’ osteogenic differentiation. Despite the identification of these key regulatory factors involved in the myofibroblastic and osteogenic differentiation of VICs, effective pharmacological therapies to prevent this pathological process are currently unavailable [9,208].

This review presents an overview of nanocarrier-based gene therapy for valvular heart disease, highlighting its potential as a novel approach in this area.

### 6.1. Therapies Targeting Valvular Interstitial Cells

The development of gene therapy utilizing nanocarriers offers a novel approach to addressing CAVD, and VICs have been the therapeutic target in several studies given their important role in the propagation phase. In a study presented by Voicu et al., C60-PEI nano-conjugates, characterized by dendrimer structures with a C60 core and branched polyethyleneimine (PEI) arms of approximately 2 kDa, were utilized as carriers for the transfection of shRNA targeting Runx2. In vitro studies showed that C60-PEI/shRNA-Runx2 nano-polyplexes significantly reduced Runx2 mRNA and protein levels, leading to decreased expression of osteogenic proteins such as alkaline phosphatase (ALP), bone sialoprotein (BSP), osteopontin (OSP), and bone morphogenetic protein 4 (BMP4) in VICs [60]. A study conjugated a collagen IV-specific peptide (Cp) to lipopolyplexes containing shRNA-Runx2 (Cp-LPP/shRunx2) to target osteoblast-differentiated VICs. The in vitro findings demonstrated that Cp-LPP/shRunx2 were effectively internalized by VICs cultured in a three-dimensional environment and resulted in a reduction in osteodifferentiation, which was evidenced by the decreased expression of osteogenic markers, lowered alkaline phosphatase activity, and reduced calcium concentration. In vivo experiments showed that Cp-LPP/shRunx2 targeted the aortic valve leaflets in a murine atherosclerosis model after retro-orbital administration, without affecting liver (AST, ALT, ALP) or kidney (urea, creatinine) function [61]. Valve fibrosis is a significant pathological characteristic of CAVD, characterized by collagen deposition within the extracellular matrix. In this study, collagen IV-targeting nanoparticles showed significantly higher accumulation in the murine heart and aorta, with increases of 18-fold and 4.7-fold, respectively, compared to non-targeting nanoparticles (both *p* < 0.01) [61]. However, this investigation primarily assessed the delivery efficiency and safety of these nanoparticles in murine models. Future research should focus on evaluating the therapeutic efficacy by measuring CAVD indicators such as peak velocity, thickness, calcification, and fibrosis areas of the aortic valves.

### 6.2. Therapies Targeting Valvular Endothelial Cells

VECs could be another possible therapeutic target for CAVD. Voicu et al. synthesized VCAM-1-targeted lipopolyplexes encapsulating shRNA-Smad3 plasmid (V-LPP/shSmad3). The lipopolyplexes effectively entered VECs and inhibited the EndoMT process in VECs subjected to HGOM by decreasing mesenchymal markers α-SMA and S100A4 and increasing the endothelial marker CD31. An in vivo study demonstrated that V-LPP/shSmad3 accumulated in the aortic root and aorta in a diabetic murine model of atherosclerosis, without negatively impacting liver or kidney function [55]. EndoMT represents another critical pathological process in CAVD, associated with the upregulation of VCAM-1 expression in VECs. Consequently, this study used a VCAM-1 recognition peptide to target these activated VECs. The study revealed that VCAM-1-targeting nanoparticles accumulated in the aorta and aortic roots at levels 17.5-fold (*p* < 0.001) and 8.4-fold (*p* < 0.01) higher, respectively, than non-targeting nanoparticles [55]. Similarly, the therapeutic efficacy of these nanoparticles should be further evaluated in vivo, too. In addition to offering direct therapeutic interventions to valvular heart diseases, gene therapy can also help improve the performance of tissue-engineered heart valves (TEHVs), which are surgically implanted to replace diseased valves. A major challenge for TEHVs is calcification, and endothelialization is considered crucial for mitigating this by minimizing platelet adhesion and covering calcified regions. In a recent study presented by Zhou et al., RunX2-siRNA and VEGF were incorporated into MSNs and immobilized onto decellularized porcine aortic valves (DPAVs) using a layer-by-layer self-assembly technique. In vitro experiments showed that the hybrid decellularized valve had excellent mechanical properties and a low hemolysis rate, and promoted endothelial cell proliferation and adhesion, while also reducing RunX2 gene expression in valve interstitial cells. In vivo experiments demonstrated that MSNs carrying RunX2-siRNA and VEGF notably improved the endothelialization of the hybrid valve [57].

### 6.3. Other Therapies

Beyond the aforementioned studies, additional research could advance gene therapy for valvular heart diseases in the future. Castro et al. conducted a study where extracellular vesicles (EVs) containing the myeloid transcription factors CCAAT enhancer binding protein alpha (CEBPA) and Spi1 were produced from human dermal fibroblasts (HDFs) after transfection with corresponding plasmids. The engineered EVs effectively transfected aortic valve cells, inducing transdifferentiation in valvular endothelial cells both in vitro and ex vivo, leading to the formation of anti-inflammatory macrophage-like cells [56]. It is an intriguing therapeutic strategy, though further investigation is required to verify its efficacy in preventing the progression of calcific aortic stenosis. In another study, Geng et al. demonstrated that alterbrassicene A bound to the P65 protein, reducing its phosphorylation and nuclear translocation in human VICs, which in turn alleviated osteogenic differentiation, as evidenced by Western blot and Alizarin red staining [58]. They further developed platelet membrane-coated nanoparticles to deliver alterbrassicene A in murine models of wire-induced aortic valve stenosis. Treatment with alterbrassicene A resulted in decreased peak velocity, calcification area, and Runx2 fluorescent intensity compared to controls [58]. This study identifies P65 as a potential target and suggests platelet membrane-coated nanoparticles as an effective delivery method for CAVD treatment.

Notably, Chen et al. developed a magnetic nanocarrier functionalized with a hexapeptide to target protease-activated receptor 2 (PAR2), according to the increased expression of PAR2 on the plasma membrane of osteogenically differentiated VICs. The study showed that magnetic nanocarriers effectively transported XCT790, an anti-calcification agent, to the calcified aortic valve using an external magnetic field. This targeted delivery inhibited VIC osteogenic differentiation, reducing aortic valve calcification and stenosis in Ldlr−/− mice on a high-fat diet [59]. This study introduces a new approach for drug delivery using magnetic nanocarriers to address the challenges of targeted delivery in calcific aortic valve disease, which are mainly due to high blood flow velocity and lack of specific biomarkers.

The studies mentioned above indicate a progressive expansion in the scope of gene therapy applications utilizing nanocarriers. It is anticipated that continued advancements in this field, particularly the integration of nanocarriers with emerging advanced technologies, will offer novel strategies for the treatment of diseased tissues that are difficult to target.

## 7. Challenges and Outlook

While the future of nanocarrier-based gene therapy appears promising, it is essential to address several critical challenges to facilitate clinical translation and ensure sustainable long-term development. These challenges include the optimization of animal models, the issue of liver accumulation, the problem of administration routes, the scalability of production processes, and the design of active targeting nanoparticles.

### 7.1. Animal Models

Current experimental animal models exhibit certain limitations. Numerous studies on CHD employ left anterior descending (LAD) coronary artery ligation in young C57/BL6 mice or Wistar rats. This approach contrasts with the clinical presentation of MI patients, who are generally older and often have multiple comorbidities influencing their prognosis. Research indicates that older animals demonstrate distinct cellular responses post-MI surgery, such as heightened cardiomyocyte apoptosis and a consequent decrease in the efficacy of caspase inhibition (anti-apoptotic) therapies [209,210]. Numerous additional variables have been identified, encompassing the mouse strain, anesthesia techniques, artery ligation methods, and even the timing of the surgical procedure [211]. In PH research, the monocrotaline and Sugen/hypoxia-induced models are commonly used animal models. However, PH is a complex and multifactorial disease, and no single animal model can accurately replicate the entire clinical spectrum of PH or each specific PH subgroup. The limitations of these animal models are further underscored by the observation that numerous drugs, which show promise in preclinical evaluations, often demonstrate limited success in subsequent clinical trials. This discrepancy highlights a translational gap commonly referred to as the “Valley of Death” [212]. Therefore, the biocompatibility, safety, and efficacy of emerging gene therapies utilizing nanocarriers require further evaluation and optimization. Organs-on-chips (OoCs), also known as microphysiological systems or tissue chips, have attracted considerable interest recently for their potential to offer valuable insights throughout the drug discovery and development process. These innovative devices can improve our comprehension of human organ function and disease mechanisms, while also providing better predictions of investigational drug safety and efficacy in humans. They are positioned to complement traditional preclinical cell culture methods and in vivo animal studies soon, with potential to replace these techniques in specific situations in the long term [213]. Based on this advanced technique, gene therapy utilizing nanocarriers can be subjected to a more precise evaluation.

### 7.2. The Accumulation in Liver

Following intravenous administration, nanoparticles predominantly accumulate in the liver as they traverse the circulatory system, with liver accumulation reported to reach up to 99% of the administered nanoparticle dose [214]. This high level of hepatic accumulation diminishes the efficiency of nanoparticle delivery, potentially leading to suboptimal therapeutic outcomes and adverse side effects. Nanoparticles accumulate significantly in the liver due to its large blood volume, slow sinusoidal flow, and direct exposure of liver macrophages to circulating nanoparticles, enhancing phagocytosis and endocytosis via multiple cellular uptake pathways [215].

Researchers have developed strategies to mitigate hepatic accumulation of nanoparticles and enhance their delivery efficiency. The first strategy focuses on nanoparticle design, which involves optimizing physicochemical parameters such as size, shape, stiffness, and surface modification. These parameters significantly affect interactions between nanoparticles and biological systems, such as serum protein adsorption, phagocytic recognition, blood vessel dynamics, endothelial adhesion, extravasation, and organ filtration. Currently, PEGylation is the most widely used method for modifying the surface of nanocarriers to decrease the liver accumulation [216]. This process forms a hydrophilic layer around nanocarriers, protecting their surface from aggregation, opsonization, and phagocytosis by the reticuloendothelial system (RES) [217]. Consequently, PEGylation extends the circulation time and enhances bioavailability. Doxil^®^, the first PEGylated nanoparticle product, received FDA approval in 1995. Doxil’s “Stealth^®^” liposomes enhanced doxorubicin’s bioavailability by nearly 90 times one week after injection compared to the free drug, with a drug half-life of 72 h and a circulation half-life of 36 h [218,219,220]. However, PEGylation is not without limitations in evading the RES. While PEGylation is generally thought to evade immune detection, recent research suggests that PEGylated nanoparticles might trigger the formation of specific anti-PEG IgM antibodies [221]. These antibodies contribute to the rapid clearance of nanoparticles from the bloodstream after repeated doses and may induce hypersensitivity reactions by activating the complement cascade [221,222]. Furthermore, prioritizing an extended circulation time does not always improve therapeutic results; thus, it is crucial to balance the stealth effect with effective interaction with diseased tissues. Utilizing a stimuli-responsive strategy to dynamically modulate the stealth effect could potentially improve functionality and augment therapeutic efficacy in the future [223,224,225].

Since most nanoparticles are captured by liver macrophages after entering the system, the second strategy focuses on reducing the phagocytic activity of these macrophages, especially Kupffer cells. This can be achieved through approaches such as Kupffer cell saturation, inhibition of their phagocytic function, and depletion of Kupffer cells [214]. Notably, the RES is crucial for removing pathogens, endotoxins, immune complexes, and other colloidal waste from the bloodstream. Interruption of the typical physiological clearance process heightens the risk of infections and other diseases, especially with repeated therapeutic administrations [226,227,228]. Therefore, it is essential to develop targeted RES block strategies and conduct further research to assess these strategies, striving to achieve an optimal balance of dynamics, safety, and efficacy in treating specific diseases [216].

### 7.3. The Route of Administration

The majority of contemporary nanomedicines are administered intravenously, which restricts their use in the management of chronic diseases. Oral medication therapy is preferred by patients due to its non-invasive, needle-free nature and lower time and cost requirements. Gastrointestinal barriers, including the gastric acid and intestinal mucosal barriers, substantially influence the bioavailability of oral medications. Key strategies to overcome gastrointestinal barriers for effective oral drug delivery involve mucus penetration, adhesion, enzyme inhibition, transient opening of tight junction proteins, and enhancing transcellular transport [229]. For example, Zhu et al. employed fluorocarbon-modified chitosan (FCS), recognized for its improved transmembrane and transmucosal penetration, to self-assemble with diverse macromolecular antibodies, creating protein complexes like FCS/IgG, FCS/*α*PD-1, and FCS/*α*CTLA4. Notably, their findings demonstrated that these complexes could transiently rearrange tight junction proteins between intestinal epithelial cells, facilitating the passage of antibodies across mucus and cellular barriers into the bloodstream [230]. Despite some unresolved issues in the practical use of oral nanomedicine, advancements in nanotechnology are anticipated to significantly enhance the clinical implementation of oral gene therapy using nanocarriers.

### 7.4. Scale-Up Production

The production methods for nanocarriers combined with gene therapies require significant improvement. Currently, most nanomedicines are synthesized in laboratory settings using various techniques that differ considerably from one case to another, often resulting in low yields. Cardiovascular diseases, which frequently necessitate prolonged treatment, highlight the economic burden associated with the limited production of nanomedicines. Therefore, achieving large-scale production is essential for the successful translation of nanomedicines into clinical practice. However, scaling up the production of nanomedicine formulations presents greater challenges compared to small-molecule drugs due to their inherent complexity. This complexity not only complicates production but also affects the reproducibility of preclinical studies involving nanomedicines. While it is relatively straightforward to control and optimize the formulation parameters of nanomedicines on a small scale, scaling up remains challenging. Minor alterations in the manufacturing process during scale-up can lead to significant changes in the physical and chemical properties of nanomedicines, thereby compromising their quality, safety, and biological efficacy in vivo [231]. Therefore, high-precision and standard methods need to be established to achieve large-scale production while enhancing the reproducibility of nanomedicine parameters and performance. Microfluidics, a technology for processing and manipulating fluid volumes from 10^−9^ to 10^−18^ L within microchannels measuring tens to hundreds of micrometers, offers a promising approach [232]. This technology facilitates accurate fluid control in microchannels, providing advantages like minimal volume, high specific surface area, and swift mass and heat transfer. These characteristics confer significant advantages, including low reagent consumption, rapid mixing, and precise control over the physicochemical properties of nanomedicines. Furthermore, microfluidics facilitates the scaling up of nanomedicine production for clinical applications through parallelization or numbering-up [232]. Microfluidic techniques have successfully scaled up the production of LNP-encapsulated mRNA vaccines like BNT162b2 and mRNA-1273, developed by Pfizer-BioNTech and Moderna for COVID-19, illustrating their feasibility in nanomedicine preparation [233,234]. Despite the persistence of several challenges, microfluidics and other emerging techniques hold promise for the scale-up production of nanomedicines.

### 7.5. Active Targeting

Nanocarrier targeting strategies for diseased tissues involve passive and active mechanisms. Passive targeting is widely used in oncology because of the “enhanced permeation and retention” (EPR) effect, which is notably significant in tumors [132]. This effect, promoting nanoparticle accumulation, is influenced by vascular factors involved in inflammation [235]. As a result, the EPR effect can also be observed in other inflammatory conditions, including arthritis, infections, and advanced atherosclerotic plaques [236,237].

Active targeting nanoparticles provide notable benefits by improving therapeutic efficacy and reducing systemic side effects compared to passive targeting. Active targeting employs affinity ligands to bind nanoparticles to antigens or extracellular matrix proteins that are uniquely or differentially overexpressed on diseased cell membranes or tissues. Active targeting nanoparticles primarily utilize ligands such as monoclonal antibodies and their fragments, proteins or protein-like molecules, peptides, nucleic acid ligands, small molecules, and sugars [238]. Our review highlights that monoclonal antibodies, peptides, and small molecules are frequently used for active targeting nanoparticles in cardiovascular diseases. Among these, monoclonal antibodies are the most extensively utilized in the creation of active targeting nanoparticles. Monoclonal antibody-targeted nanoparticles encounter major obstacles due to their large molecular size, instability in organic solvents, and potential immunogenicity, which may result in rapid clearance and complicate nanoparticle scale-up and manufacturing processes [239,240]. To address these challenges, antigen-binding antibody fragments have been created [241]. Peptides and small molecules are attractive as targeting ligands due to their compact size, low immunogenicity, stability, ease of conjugation, potential for high ligand density on nanoparticle surfaces, and scalable manufacturing processes [37,242,243]. These attributes render them more suitable as ligands for active targeting.

Beyond the intrinsic traits of targeting ligands, the physicochemical properties of nanoparticles—such as size, shape, surface charge, surface chemistry, hydrophobicity, roughness, rigidity, and compositional complexity—also affect their uptake and targeting to specific organs, tissues, or cells [244]. Therefore, careful consideration of these factors is essential in designing optimal nanoparticles for effective active targeting.

## 8. Conclusions

Many diseases in the cardiovascular system are often complex and heterogeneous, characterized by diverse gene expression patterns that influence their progression and treatment response. Heterozygous loss-of-function mutations in PAH-related genes, such as BMPR2, ACVRL1, ENG, GDF2, BMP10, and KCNK3, have been associated with pathogenic features of PAH, including depolarization, proliferation, apoptosis resistance, and PASMC constriction [11,187]. Genome-wide association studies (GWAS) in European ancestry populations have identified risk loci such as ALPL, IL6, LPA, and palmdelphin (PALMD), which are significantly linked to CAVD [9]. Additionally, bicuspid aortic valve (BAV) malformation, the most prevalent congenital cardiac defect that often progresses to CAVD, exhibits high heritability (approximately 47–89%). A genetic association study with 1326 BAV cases and 4660 controls identified two significant risk loci: one in DHX38 and another in an intergenic region on chromosome 8 near CTSB and DEFB135 [9]. Similarly, numerous genes associated with coronary artery disease and hypertension have been identified so far [4,10]. However, only a limited number of these genes, such as PCSK9, have been translated into clinical practice, and most require further investigation to assess their roles in pathogenesis and therapeutic efficacy. With recent technological advancements, gene therapy based on nanocarriers may offer targeted treatments tailored to the specific etiology of individual patients owing to precisely genetic diagnostics in the future.

In summary, nanocarrier-based gene therapy offers a promising strategy for treating cardiovascular diseases, despite some unresolved challenges. Progress in complementary technologies like organ-on-chips and microfluidics is anticipated to drive advancements in this therapeutic field soon.

## Figures and Tables

**Figure 1 ijms-26-01743-f001:**
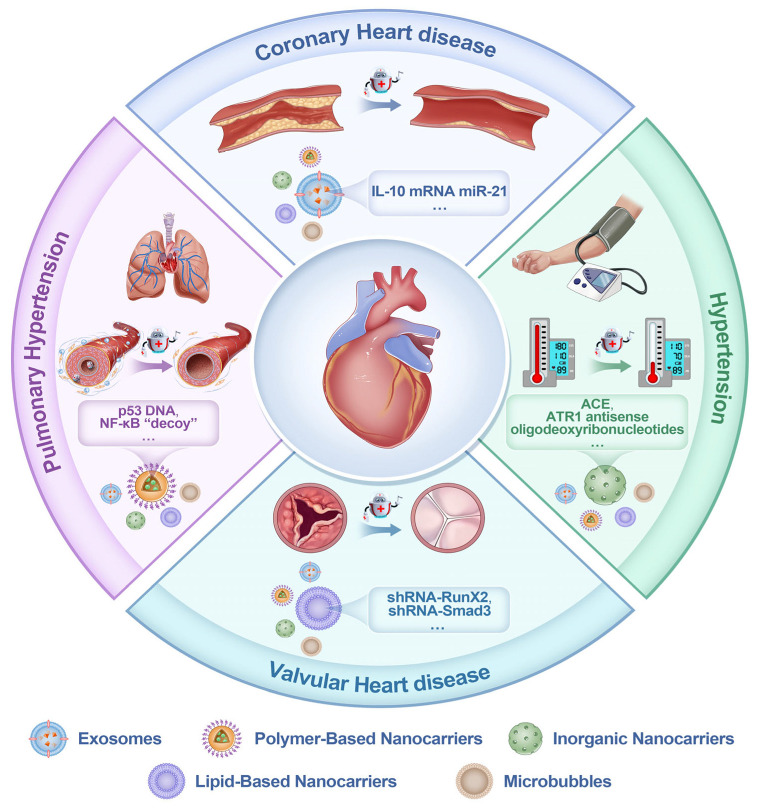
Examples of gene therapy based on nanocarriers in cardiovascular diseases.

**Figure 2 ijms-26-01743-f002:**
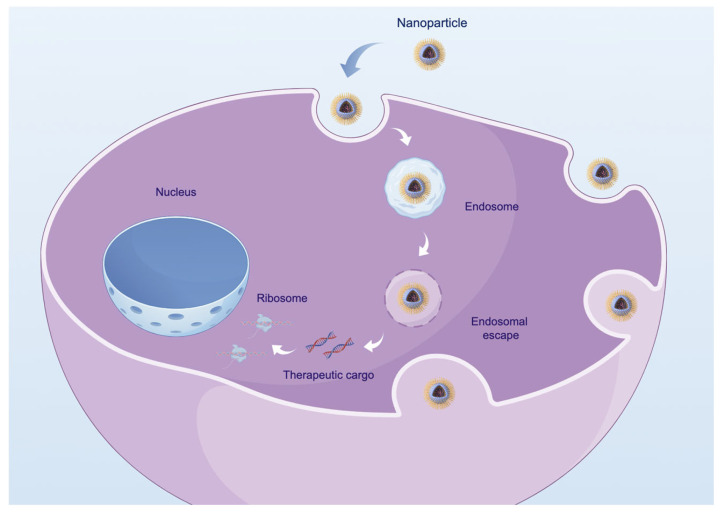
The process of nanocarriers entering the cell (By Figdraw).

**Figure 4 ijms-26-01743-f004:**
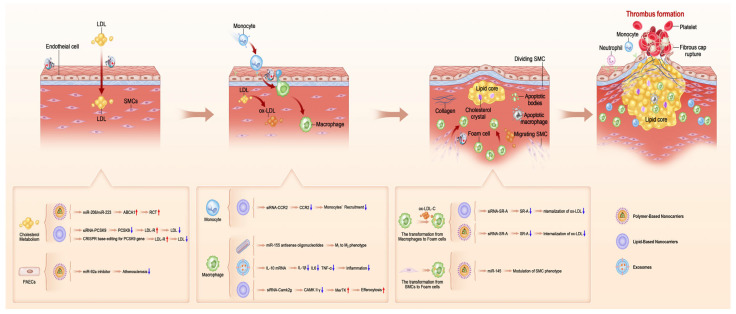
The pathological progress of atherosclerosis and possible therapeutic targets (original creation).

**Figure 5 ijms-26-01743-f005:**
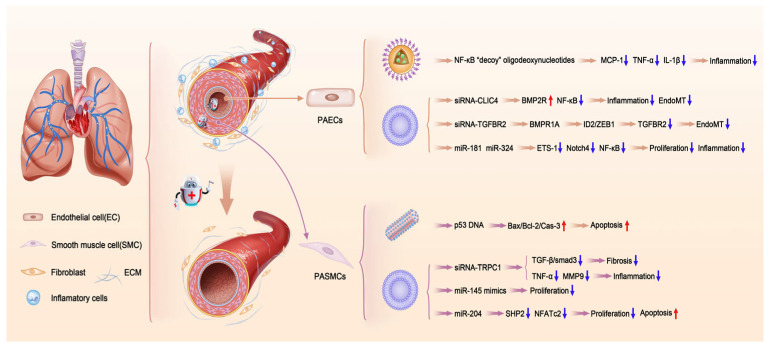
Gene therapies based on nanocarriers in PAH.

**Table 1 ijms-26-01743-t001:** The summary of all studies based on nanocarriers in this review.

Disease	Nanocarriers	Therapeutic Target	Therapeutic Cargo	Active Targeting Strategy	Model	Findings	Ref
CHD-Atherosclerosis	Polymer-based nanocarriers	Endothelial cells	miR-92a	VCAM-1-targeting peptide, VHPKQHR	Apolipoprotein E-deficient (ApoE−/−) mouse model	An 83% reduction in atherosclerotic lesions	[21]
CHD-Atherosclerosis/Myocardial infarction	Lipid-based nanocarriers	Monocytes	siRNA-CCR2	-	Apolipoprotein E-deficient (ApoE−/−) mouse model/Myocardial infarction mouse model	A 38% reduction in atherosclerotic lesion size and a 34% reduction myocardial infarct size	[22]
CHD-Atherosclerosis	Baicalein nanorods	Macrophages	antisense oligonucleotides targeting miR-155	sialic acid	In vitro model and rat model fed with high-fat diets	A phenotypic shift in the macrophages; attenuated inflammation; increased artery lumen diameter	[23]
CHD-Atherosclerosis	Exosomes	Macrophages	IL-10 mRNA	-	Apolipoprotein E-deficient (ApoE−/−) mouse model	Attenuated inflammation and reduced atherosclerotic lesion size	[24]
CHD-Atherosclerosis	Lipid-based nanocarriers	Macrophages	siRNA-Camk2g	peptide ligand S2p	A Western diet-fed low-density lipoprotein receptor-deficient (Ldlr−/−) mouse model	Improved efferocytosis, decreased necrotic core area, and increased fibrous cap thickness	[25]
CHD-Atherosclerosis	HDL-based nanocarriers	Macrophages	siRNA-SR-A + pitavastatin	CD36 ligand phosphatidylserine	Apolipoprotein E-deficient (ApoE−/−) mouse model	A 65.8% reduction in atherosclerotic lesions	[26]
CHD-Atherosclerosis	Polymer-based nanocarriers	Smooth muscle cells	miR-145 mimics	MCP-1 peptide	Apolipoprotein E-deficient (ApoE−/−) mouse model fed with Western diets	A reduction in lesion size (49 ± 0.1%) and in necrotic core area (73 ± 0.3%)	[27]
CHD-Atherosclerosis	Polymer-based nanocarriers	Cholesterol metabolism	miR-206/miR-233 mimics	-	In vitro and C57BL6 wild-type mice model	Enhanced reverse cholesterol transport	[28]
CHD-Atherosclerosis	HDL-based nanocarriers	Cholesterol metabolism	antagomiR-33a	-	In vitro model	Promoted cholesterol efflux in foam cells	[29]
CHD-Atherosclerosis	Polymer-based nanocarriers	Macrophages and cholesterol metabolism	siRNA-SR-A and ligands for LXRs	mannose	A Western diet-fed low-density lipoprotein receptor-deficient (Ldlr−/−) mouse model	A reduction in total (29.9 ± 7.1 vs. 17.4 ± 3.7) and aortic arch (54.6 ± 8.6 vs. 36.2 ± 8.1) plaque area	[30]
CHD-Atherosclerosis	Lipid-based nanocarriers	Cholesterol metabolism	siRNA-PCSK9	-	Rodent and nonhuman primate models	Decreased level of cholesterol	[31]
CHD-Atherosclerosis	Lipid-based nanocarriers	Cholesterol metabolism	CRISPR base editing for PCSK9	-	Mouse and nonhuman primate models	Decreased level of cholesterol	[32]
CHD-Myocardial infarction	Exosomes	Macrophages and cardiomyocytes	miR-98-5p	-	Myocardial infarction rat model	Attenuated inflammation and oxidative stress; reduced infarct size	[33]
CHD-Myocardial infarction	Exosomes	Macrophages	miR-182	-	In vitro and myocardial infarction mouse model	A phenotypic shift in the macrophages; attenuated inflammation and decreased infarct size	[34]
CHD-Myocardial infarction	Exosomes	Macrophages	miR-21-5p	-	In vitro and myocardial infarction mouse model	Reduced expression of inflammatory factors	[35]
CHD-Myocardial infarction	Polymer-based nanocarriers	Macrophages	miRNA-21 mimics	-	Myocardial infarction mouse model	A phenotypic shift in the macrophages; mitigated hypertrophy, fibrosis, and cellular apoptosis	[36]
CHD-Myocardial infarction	Polymer-based nanocarriers	Endothelial cells	siRNA-VCAM-1 + dexamethasone	cyclic arginine-glycine-aspartic acid (cRGD)	Myocardial infarction rat model	Attenuated inflammation; reduced infarct size (37% reduction), cardiac fibrosis, and cardiomyocyte apoptosis; restored systolic function	[37]
CHD-Myocardial infarction	Microbubbles	Cardiomyocytes	human SDF-1α-nuclear factor κB plasmid	ultrasound-targeted microbubble destruction	Myocardial infarction rabbit model	Promoted angiogenesis, reduced myocardial fibrosis, preserved cardiac function, and decreased infarct size (about 7.2% reduction)	[38]
CHD-Myocardial infarction	Inorganic nanocarriers (MSNs)	Cardiomyocytes	siRNA-RAGE + dexamethasone	PPTP (ROS-responsive)	Myocardial infarction rat model	Reduced myocardial fibrosis and apoptosis, and restored systolic function	[39]
CHD-Myocardial infarction	Lipid-based nanocarriers	Cardiomyocytes	anti-miR-1 antisense oligonucleotides	anti-cardiac troponin I (cTnI) antibody	Myocardial infarction rat model	Restored depolarized resting membrane potential and alleviated ischemic arrhythmias	[40]
CHD-Myocardial infarction	Lipid-based nanocarriers	Cardiomyocytes	miR-21	anti-cardiac troponin T (cTnT) antibody	Myocardial infarction rat model	Improved the cardiac function and decreased infarct size (21.9% reduction)	[41]
CHD-Myocardial infarction	Polymer-based nanocarriers	Endothelial cells and cardiomyocytes	miR-133	arginine-glycine-aspartic acid (RGD)	Myocardial infarction rat model	Suppressed cardiomyocyte apoptosis, inflammation, and oxidative stress; preserved cardiac structure and function	[42]
CHD-Myocardial infarction	Exosomes	Endothelial cells and cardiomyocytes	miR-221-3p	-	In vitro and myocardial infarction rat model	Promoted angiogenesis, reduced apoptosis, and preserved cardiac function	[43]
CHD-Myocardial infarction	Exosomes	Macrophages, fibroblasts, and endothelial cells	miR-125a-5p	-	Myocardial infarction mouse model	Improved myocardial contractile function and reduced cardiac remodeling	[44]
Pulmonary hypertension	Lipid-based nanocarriers	PASMCs	synthetic miR-204 mimics	-	MCT-induced rat model	Reduced wall thickness of pulmonary arteries and reduced mPAP	[45]
Pulmonary hypertension	Lipid-based nanocarriers	PASMCs	an antisense oligonucleotide against miR-145	-	Sugen/hypoxia-induced rat model	Reduced wall thickness of pulmonary arteries and reduced density of occlusive vascular lesions	[46]
Pulmonary hypertension	Lipid-based nanocarriers	PASMCs	siRNA-TRPC1	-	Hypoxia-induced mouse model	Attenuated right ventricle and pulmonary arteriolar remodeling, and reduced RVSP	[47]
Pulmonary hypertension	Baicalein nanocrystals	PASMCs	pDNA of p53	glucuronic acid for targeting GLUT-1	MCT-induced rat model	Suppressed proliferation of PASMCs and reduced mPAP	[48]
Pulmonary hypertension	Lipid-based nanocarriers	PAECs	miR-181a-5 and miR-324-5p	-	Sugen/hypoxia-induced mouse model	Reduced pulmonary muscularization and reduced RVSP	[49]
Pulmonary hypertension	Polymer-based nanocarriers	PAECs	siRNA-TGFBR2	-	BMPR1a^iECKO^ mouse model	Reduced pulmonary muscularization and reduced RVSP	[50]
Pulmonary hypertension	Lipid-based nanocarriers	PAECs	siRNA-CLIC4	-	Sugen/hypoxia-induced rat model	Reduced pulmonary muscularization and reduced RVSP	[51]
Pulmonary hypertension	Polymer-based nanocarriers	PAECs	NF-κB decoy	-	MCT-induced rat model	Attenuated inflammation and small pulmonary arterial remodeling, reduced RVSP, and improved the survival rate	[52]
Hypertension	Inorganic nanocarriers	ACE	ACE gene-targeted antisense oligonucleotide	-	Inherited stress-induced arterial hypertension rat model	Reduction in systolic blood pressure (20–30 mmHg)	[53]
Hypertension	Hybrid nanocarriers (silicon–organic)	ACE and AT1R	antisense oligodeoxyribonucleotides targeting the mRNA of ACE and AT1R	-	Inherited stress-induced arterial hypertension rat model	Reduction in systolic blood pressure (approximately 30 mmHg)	[54]
Valvular heart disease-CAVD	Hybrid nanocarriers (lipid–polymer)	VECs	shRNA-Smad3	VCAM-1 recognition peptide	In vitro model and a mouse model of atherosclerosis complicated by diabetes	Mitigated EndoMT process and the accumulation of nanocarriers in the aortic root and aorta	[55]
Valvular heart disease-CAVD	Exosomes	VECs	CEBPA and Spi1	-	In vitro and ex vivo models	The generation of anti-inflammatory macrophage-like cells	[56]
Valvular heart disease-CAVD	Inorganic nanocarriers (MSNs)	VECs and VICs	siRNA-RunX2 and VEGF	-	In vitro and in vivo models	Enhanced the endothelialization of TEHV	[57]
Valvular heart disease-CAVD	Cell membrane-coated nanocarriers	VICs	alterbrassicene A	platelet membrane	In vitro and a wire-induced aortic valve stenosis mouse model	Reduced peak velocity and deposition of calcium	[58]
Valvular heart disease-CAVD	Hybrid nanocarriers (polymer–inorganic)	VICs	XCT790	a hexapeptide targeting PAR2 combined with external magnetic field	In vitro and a high-fat diet-fed low-density lipoprotein receptor-deficient (Ldlr−/−) mouse model	Reduced peak velocity, thickness of valves, and deposition of collagen and calcium	[59]
Valvular heart disease-CAVD	Hybrid nanocarriers (polymer–inorganic)	VICs	shRNA-Runx2	-	In vitro model	Mitigated osteogenic differentiation	[60]
Valvular heart disease-CAVD	Hybrid nanocarriers (lipid–polymer)	VICs	shRNA-Runx2	a peptide with specific affinity for collagen IV	In vitro model and a mouse model of atherosclerosis	Mitigated osteogenic differentiation and the accumulation of nanocarriers in the aortic valve leaflets	[61]

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
