# Peer review of "Emerging Gene Therapy Based on Nanocarriers: A Promising Therapeutic Alternative for Cardiovascular Diseases and a Novel Strategy in Valvular Heart Disease"

_ijms, 2025, doi:10.3390/ijms26041743_

Round 1

Reviewer 1 Report

Comments and Suggestions for Authors

1. What is the main question addressed by the research?

                     The introduction lacks information about the delivery mechanism of nanocarriers in the human body; this information is needed to support the objective and research question of the study.

                     Include a section discussing the delivery mechanism of these therapies in the human body to complement the study.

2. Do you consider the topic original or relevant to the field? Does it address a specific gap in the field?

In line 87, one of the aims mentions that this is the first review summarizing gene therapy based on nanocarriers in the context of valvular heart disease. However, if the authors wish to emphasize this novel aspect more prominently, they may consider modifying the title to better reflect this unique contribution and giving Section 6 more relevance and prominence.

3. What does it add to the subject area compared with other published material?

Place more emphasis on the novelty of this therapy in valvular heart disease

4. Are the conclusions consistent with the evidence and arguments presented and do they address the main question posed?

Create a specific conclusion section: 8. Conclusion and provide more argumentation in the conclusion. Although it meets the requirements, it is too brief for a review article. Additionally, you could include perspectives within the conclusion.

5. Are the references appropriate?

In the bibliography, check that it follows the author's guidelines, as some journal names are not abbreviated.

6. Any additional comments on the tables and figures.

Add a table with relevant studies on in vivo and in vitro research.

Detail comments

1.      The article adequately addresses the topics discussed; however, it is necessary to standardize the use of abbreviations. Although a list of abbreviations is provided for reference, some sections of the text define the abbreviations the first time they are mentioned, while other sections repeat these definitions despite having already introduced them. It is essential to carefully review where each abbreviation is first mentioned and remove redundant definitions. You should standardize their use, ensuring that each abbreviation is defined only once within the text, with subsequent mentions using only the abbreviated form.

2.      The figures are described only in captions, but no reference is made to them within the main text. Since figures are used to explain important content visually, it is essential to explicitly mention them in the text where relevant to enhance the reader's understanding.

3.      Additionally, it is necessary to cite the source of the figures if they were taken from other works or indicate that they are original creations if applicable. It is also recommended that the resolution of the text within the figures be improved to ensure readability and visual quality.

4.      Add a citation on line 30 for this sentence: Cardiovascular disease constitutes a primary cause of mortality worldwide.

5.      Line 63: provide the full name of the abbreviation PKP2.

6.      Line 80: provide the full name of the abbreviation f PCSK9 and LDL-C.

7.      Line 97: Add a reference at the end of the sentence.

8.      Line 77: Add a reference.

9.      Line 226: provide the full name of the abbreviation CCR2.

10. Line 238: provide the full name of the abbreviation mTORC1.

11. Line 242: provide the full name of the abbreviation IL-10.

12. Line 252: provide the full name of the abbreviation LXRα.

13. Line 261: provide the full name of the abbreviation Camk2g.

14. Line 265: provide the full name of the abbreviation Ldlr.

15. Line 272: provide the full name of the abbreviation CD3.

16. Line 275: provide the full name of the abbreviation Nf-kB.

17. Line 285: Use "MicroRNA-145 (miR-145)" consistently throughout the text.

18. Line 288: provide the full name of the abbreviation DSPE-PEG.

19. Line 289: provide the full name of the abbreviation MCP-1.

20. Line 293: provide the full name of the abbreviation PBS.

21. Line 298: provide the full name of the abbreviation ABCA1 and ABCG2

22. Line 327: provide the full name of the abbreviation LDL-C.

23. Line 359: provide the full name of the abbreviation CAD.

24. Line 377: provide the full name of the abbreviation RPPT.

25. Line 379: provide the full name of the abbreviation cRGD and DXM.

26. Line 380: provide the full name of the abbreviation siVCAM-1.

27. Line 392 and 393: Change "nanoparticles" to "NPs."

28. Line 414: Remove "MicroRNAs" and keep only "miRNAs," as previously defined.

29. Lines 439, 446, 454, and throughout the rest of the text: Standardize "MicroRNAs" or "miRNAs" consistently.

30. Line 442: provide the full name of the abbreviation RGD.

31. Line 452: provide the full name of the abbreviation cT-21-LIPs.

32. Lines 451 and 456: Standardize the usage of "anti-cTnT antibody" or "anti-cardiac troponin I (cTnI) antibody" consistently throughout the text. Choose one term and use it uniformly.

33. Line 456: provide the full name of the abbreviation AMO-1.

34. Line 295: provide the full name of the abbreviation CCL2.

35. Line 530: provide the full name of the abbreviation BMPRII.

36. Line 531: provide the full name of the abbreviation PPAR- γ and FOXO1.

37. Line 532: provide the full name of the abbreviation HIF-1α.

38. Line 551: Change "are" to "were."

39. Line 560: provide the full name of the abbreviation DACC.

40. Line 570: provide the full name of the abbreviation TGFBR2.

41. Line 571: provide the full name of the abbreviation 7C1.

42. Line 573: provide the full name of the abbreviation BMPR1aiECKO.

43. Line 576: provide the full name of the abbreviation ETS1.

44. Line 589: provide the full name of the abbreviation MCT.

45. Line 650: provide the full name of the abbreviation ISIAH.

46. Line 684: Remove "etc.".

47. Line 710: provide the full name of the abbreviation ALP, BSP, OSP, BMP4.

48. Line 745: provide the full name of the abbreviation CEBPA.

49. Line 743: You mention several studies, but further down, you only cite two. Could you include a table with more studies you have found?

50. Line 779: provide the full name of the abbreviation PALMD.

51. Line 803: provide the full name of the abbreviation PH.

Author Response

  1. The manuscript adequately addresses the topics discussed; however, it is necessary to standardize the use of abbreviations. Although a list of abbreviations is provided for reference, some sections of the text define the abbreviations the first time they are mentioned, while other sections repeat these definitions despite having already introduced them. It is essential to carefully review where each abbreviation is first mentioned and remove redundant definitions. You should standardize their use, ensuring that each abbreviation is defined only once within the text, with subsequent mentions using only the abbreviated form.

Response: We thank the reviewer for this suggestion and the abbreviations in this manuscript have been further standardized.

  1. The figures are described only in captions, but no reference is made to them within the main text. Since figures are used to explain important content visually, it is essential to explicitly mention them in the text where relevant to enhance the reader's understanding.

Response: We thank the reviewer for this suggestion and the references of all the figures has been made within the main text.

  1. Additionally, it is necessary to cite the source of the figures if they were taken from other works or indicate that they are original creations if applicable. It is also recommended that the resolution of the text within the figures be improved to ensure readability and visual quality.

Response: We thank the reviewer for this suggestion. The Figure 1, 2, 4 and 5 are original creations and we have marked them in the manuscript. Meanwhile, Figure 3 are taken from other articles and the references have been cited.

  1. Add a citation on line 30 for this sentence: Cardiovascular disease constitutes a primary cause of mortality worldwide.

Response: The reference has been added in this line.

  1. Line 63: provide the full name of the abbreviation PKP2.

Response: The full name of PKP2 has been added in the manuscript as “plakophilin 2”.

  1. Line 80: provide the full name of the abbreviation PCSK9 and LDL-C.

Response: The full names of PCSK9 and LDL-C have been added in the manuscript as “proprotein convertase subtilisin/kexin type 9” and “low-density lipoprotein cholesterol”, respectively.

  1. Line 97: Add a reference at the end of the sentence.

Response: The reference has been added.

  1. Line 77: Add a reference.

Response: The reference has been added.

  1. Line 226: provide the full name of the abbreviation CCR2.

Response: The full name of CCR2 has been added in the manuscript as “C-C chemokine receptor 2”.

  1. Line 238: provide the full name of the abbreviation mTORC1.

Response: The full name of mTORC1 has been added in the manuscript as “mechanistic target of rapamycin complex 1”.

  1. Line 242: provide the full name of the abbreviation IL-10.

Response: The full name of IL-10 has been added in the manuscript as “Interleukin-10”.

  1. Line 252: provide the full name of the abbreviation LXRα.

Response: The full name of LXRα has been added in the manuscript as “Liver X receptor-alpha”.

  1. Line 261: provide the full name of the abbreviation Camk2g.

Response: The full name of Camk2g is the Ca2+/calmodulin-dependent protein kinase γ,which is the same as CaMKIIγ. Actually, this form of “Camk2g” is used to represent the gene name of Ca2+/calmodulin-dependent protein kinase γ.

  1. Line 265: provide the full name of the abbreviation Ldlr.

Response: The full name of Ldlr is low-density lipoprotein receptor , which is the same as LDL-R. Similarly, this form of “Ldlr” is used to represent the gene name of low-density lipoprotein receptor.

  1. Line 272: provide the full name of the abbreviation CD36.

Response: The full name of CD36 has been added in the manuscript as “Cluster of Differentiation 36”.

  1. Line 275: provide the full name of the abbreviation NF-κB.

Response: The full name of NF-κB has been added in the manuscript as “nuclear factor-kappaB”.

  1. Line 285: Use "MicroRNA-145 (miR-145)" consistently throughout the text.

Response: We have changed “MicroRNA-145” to “ miRNA-145” as the abbreviation of “ miRNA” has been used in the earlier section of this manuscript. And the abbreviation miR-145 has been used consistently throughout the text.

  1. Line 288: provide the full name of the abbreviation DSPE-PEG(2000).

Response: The full name of DSPE-PEG(2000) has been added in the manuscript as “1,2-distearoyl-sn-glycero-3-phosphoethanolamine-N-[(polyethylene glycol)-2000]”.

  1. Line 289: provide the full name of the abbreviation MCP-1.

Response: The full name of MCP-1 has been added in the manuscript as “monocyte chemoattractant protein-1 ”.

  1. Line 293: provide the full name of the abbreviation PBS.

Response: The full name of PBS has been added in the manuscript as “phosphate-buffered saline ”.

  1. Line 298: provide the full name of the abbreviation ABCA1 and ABCG2

Response: The full names of ABCA1 and ABCG2 have been added in the manuscript as “ATP-binding cassette A1 ” and “ATP-binding cassette transporter G2 ”, respectively.

  1. Line 327: provide the full name of the abbreviation LDL-C.

Response: The full name of LDL-C has been added in the manuscript as “low-density lipoprotein cholesterol” in line 48,where it was mentioned the first time.

  1. Line 359: provide the full name of the abbreviation CAD.

Response: Coronary artery disease (CAD) is another name of coronary heart disease (CHD). We have changed it to CHD to make it consistent with the earlier section of the manuscript.

  1. Line 377: provide the full name of the abbreviation RPPT.

Response: The full name of RPPT has been added in the manuscript as “cRGD-modified, PEGylated, ditellurium-crosslinked polyethylenimine”.

  1. Line 379: provide the full name of the abbreviation cRGD and DXM.

Response: The full names of cRGD and DXM have been added in the manuscript as “cyclic arginine-glycine-aspartic acid ” and “dexamethasone ”, respectively. But we further changed DXM to Dex to make it consistent with the earlier section of the manuscript.

  1. Line 380: provide the full name of the abbreviation siVCAM-1.

Response: We have changed “siVCAM-1” to “VCAM-1 siRNA” to make it more readily comprehensible and the full name of VCAM-1 has been added in the earlier section of the manuscript.

  1. Line 392 and 393: Change "nanoparticles" to "NPs."

Response: The term "nanoparticles" is frequently employed in the manuscript; however, in certain instances, the use of its abbreviation is inappropriate. Consequently, we have removed the abbreviation and consistently used the full term throughout the manuscript.

  1. Line 414: Remove "MicroRNAs" and keep only "miRNAs," as previously defined.

Response: "MicroRNAs" in this line has been removed.

  1. Lines 439, 446, 454, and throughout the rest of the text: Standardize "MicroRNAs" or "miRNAs" consistently.

Response: We have standardized “miRNAs” consistently throughout the manuscript.

  1. Line 442: provide the full name of the abbreviation RGD.

Response: The full name of RGD has been added in the manuscript as “arginine-glycine-aspartic acid”.

  1. Line 452: provide the full name of the abbreviation cT-21-LIPs.

Response: cT-21-LIPs is not an abbreviation but the name of the nanoparticle developed by Li et al. We have made this statement more clear in line 354-355.

  1. Lines 451 and 456: Standardize the usage of "anti-cTnT antibody" or "anti-cardiac troponin I (cTnI) antibody" consistently throughout the text. Choose one term and use it uniformly.

Response: cTnT and cTnI are different myocardium specific troponins and the full name of cTnT has been added in the manuscript as “cardiac troponin T”.

  1. Line 456: provide the full name of the abbreviation AMO-1.

Response: The full name of AMO-1 has been added in the manuscript as “anti-miR-1 antisense oligonucleotides”.

  1. Line 295: provide the full name of the abbreviation CCL2.

Response: The full name of CCL2 has been added in the manuscript as “C-C motif chemokine ligand 2”.

  1. Line 530: provide the full name of the abbreviation BMPRII.

Response: The full name of BMPRII has been added in the manuscript as “bone morphogenic protein type II receptor”.

  1. Line 531: provide the full name of the abbreviation PPAR-γ and FOXO1.

Response: The full names of PPAR-γ and FOXO1 have been added in the manuscript as “peroxisome proliferator-activated receptor gamma” and “Forkhead box protein O1 ”, respectively.

  1. Line 532: provide the full name of the abbreviation HIF-1α.

Response: The full name of HIF-1α has been added in the manuscript as “hypoxia-inducible factor-1 alpha”.

  1. Line 551: Change "are" to "were."

Response: The word "are" has been changed to "were" in this line.

  1. Line 560: provide the full name of the abbreviation DACC.

Response: DACC is actually not an abbreviation but the full name of the novel siRNA delivery system developed by Fehring et. al.. We have added more details about this nanocarrier in line 418.

  1. Line 570: provide the full name of the abbreviation TGFBR2.

Response: The full name of TGFBR2 has been added in the manuscript as “transforming growth factor-beta type II receptor”.

  1. Line 571: provide the full name of the abbreviation 7C1.

Response: 7C1 is actually not an abbreviation but the full name of the nanocarrier. We have added more details about this nanocarrier in line 426 to 427.

  1. Line 573: provide the full name of the abbreviation BMPR1aiECKO.

Response: The full name of BMPR1aiECKO has been added in the manuscript as “endothelial deletion of Bmpr1a”.

  1. Line 576: provide the full name of the abbreviation ETS-1.

Response: The full name of ETS-1 has been added in the manuscript as “E26 transformation-specific sequence-1”.

  1. Line 589: provide the full name of the abbreviation MCT.

Response: The full name of MCT has been added in the manuscript as “monocrotaline”.

  1. Line 650: provide the full name of the abbreviation ISIAH.

Response: The full name of ISIAH has been added in the manuscript as “inherited stress-induced arterial hypertension”.

  1. Line 684: Remove "etc.".

Response: The word "etc." has been removed in this line.

  1. Line 710: provide the full name of the abbreviation ALP, BSP, OSP, BMP4.

Response: The full names of the abbreviation ALP, BSP, OSP, BMP4 have been added in the manuscript as “alkaline phosphatase” , “bone sialoprotein” , “osteopontin” and “bone morphogenetic protein 4”, respectively.

  1. Line 745: provide the full name of the abbreviation CEBPA.

Response: The full name of CEBPA has been added in the manuscript as “CCAAT enhancer binding protein alpha”.

  1. Line 743: You mention several studies, but further down, you only cite two. Could you include a table with more studies you have found?

Response: We thank the reviewer for this suggestion. Research on gene therapy based on nanocarriers for the treatment of valvular heart disease remains limited. We have added an additional study to this part (line 536 to 541 ), and these represent the entirety of studies currently accessible on the website. A comprehensive summary of these studies is further provided in Table 1.

  1. Line 779: provide the full name of the abbreviation PALMD.

Response: The full name of PALMD has been added in the manuscript as “palmdelphin”.

  1. Line 803: provide the full name of the abbreviation PH.

Response: PH is the abbreviation of pulmonary hypertension and this abbreviation has been added in line 374, where it was mentioned the first time.

  1. Other Comments.

Response: We have modified the title and enriched the content of Section 6 to give this part more prominence. And the Section of Conclusion has also been adjusted, too. Furthermore, we have summarized the details of all relevant studies in Table 1.

Reviewer 2 Report

Comments and Suggestions for Authors

This review focuses on the emerging nanocarrier gene therapy. Given the rapid development of gene therapy and nanotechnology, this topic is of great cutting-edge significance and importance. It provides new ideas and directions for the treatment of cardiovascular diseases and offers crucial guidance and inspiration for both clinical practice and fundamental research. However, there are still some issues that need to be revised before formal acceptance.

1.The article primarily covers cardiovascular diseases such as coronary heart disease, pulmonary hypertension, hypertension, and valvular heart disease. Nevertheless, only a simple schematic diagram (Figure 1) of the application of gene therapy based on nanocarriers in cardiovascular diseases is provided, failing to fully utilize the role of diagrams in a review.

2.The article elaborately describes the applications of nanocarriers in different pathological stages, including treatment strategies targeting key aspects such as endothelial cells, macrophages, foam cell formation, and cholesterol metabolism. In the section on coronary heart disease, numerous research examples are cited, such as the targeted delivery of miR - 92a inhibitor (detailed in Section 3.1.1) and the encapsulation of CCR2 - siRNA (mentioned in Section 3.1.2). However, when presenting the effects of nanocarrier therapies, only simple data like the percentage reduction of lesions are listed. There is no analysis of the statistical significance of the data, nor is there an exploration of the causes of data fluctuations and their impact on the reliability of conclusions. Detailed statistical analysis needs to be supplemented.

3.In the section introducing the classification of nanocarriers (2. Categories for Nanocarriers), the descriptions of the synthesis methods and chemical structures of various nanocarriers are relatively brief. For instance, the synthesis principles and structural characteristics of commonly used polymers such as PLA, PGA, and PLGA are not further elaborated. Only their applications in drug delivery are mentioned, which may make it difficult for readers unfamiliar with this field to deeply understand the characteristics and advantages of different nanocarriers.

4.Please supplement content related to the synthesis details and structural properties of various nanocarriers. You can cite some classic research literature on preparation methods and illustrate with simple chemical structure diagrams. When describing the application of each nanocarrier in cardiovascular diseases, further explore the relationship between its action mechanism and the structure and properties of the nanocarrier, that is, why a specific nanocarrier is suitable for a specific treatment target.

5.The author can add more detailed analyses of actual clinical cases, including the impact of individual patient differences (such as age, gender, comorbidities, etc.) on treatment outcomes. Meanwhile, introduce analyses of some failure cases to explore possible causes of failure, such as the immunogenicity of nanocarriers and the instability of drug release, to make the analysis more comprehensive.

6.The article mentions the clinical trial progress of some gene therapy products, such as VERVE - 101, etc. However, the discussion of the key clinical information of these trials is not comprehensive enough. For example, in the section on hypertension treatment, the nanocarrier gene therapy is not comprehensively compared with traditional thiazide diuretics, angiotensin - converting enzyme (ACE) inhibitors, and calcium channel blockers (CCBs), limiting the guiding value of the article for clinical practice.

7.On the existing basis, add multiple types of diagrams. For example, draw schematic diagrams of the structures of nanocarriers to show their action processes within cells; create diagrams that associate disease pathological processes with the treatment targets of nanocarriers to more clearly present the treatment mechanisms.

8.It is recommended to use bar charts or line charts to compare the treatment effects of different nanocarriers in cardiovascular diseases, visually demonstrating data differences.

9.There are no relevant references cited regarding the water solubility and first - pass effect of antihypertensive drugs in lines 40 - 42.

10.The article is clearly structured, proceeding in sequence according to the classification of nanocarriers (2. Categories for Nanocarriers), their applications in different cardiovascular diseases (Sections 3 - 6 discuss different diseases respectively), and challenges and prospects (7. Challenges and Outlook). However, in the discussion of each disease, the pathological processes and corresponding treatment strategies are further subdivided. For the complex classification of nanocarriers, action mechanisms, and disease pathological processes, the content in the article may not be clearly explained. Consider adding more detailed diagrams to help readers better understand.

11.The expressions of some professional terms in the article are inconsistent, affecting readers' understanding of the content. Standardize the use of professional terms throughout the article. Provide a complete definition and explanation when a term first appears and maintain consistency thereafter.

12.Some sentences have complex structures and obscure expressions, reducing the readability of the article. Please simplify and optimize complex sentences.

Comments on the Quality of English Language

can be improved.

Author Response

  1. The article primarily covers cardiovascular diseases such as coronary heart disease, pulmonary hypertension, hypertension, and valvular heart disease. Nevertheless, only a simple schematic diagram (Figure 1) of the application of gene therapy based on nanocarriers in cardiovascular diseases is provided, failing to fully utilize the role of diagrams in a review.

Response: We thank the reviewer for this point. After this revision, one table and five figures have been used in this review.

  1. The article elaborately describes the applications of nanocarriers in different pathological stages, including treatment strategies targeting key aspects such as endothelial cells, macrophages, foam cell formation, and cholesterol metabolism. In the section on coronary heart disease, numerous research examples are cited, such as the targeted delivery of miR - 92a inhibitor (detailed in Section 3.1.1) and the encapsulation of CCR2 - siRNA (mentioned in Section 3.1.2). However, when presenting the effects of nanocarrier therapies, only simple data like the percentage reduction of lesions are listed. There is no analysis of the statistical significance of the data, nor is there an exploration of the causes of data fluctuations and their impact on the reliability of conclusions. Detailed statistical analysis needs to be supplemented.

Response: We thank the reviewer for this suggestion and detailed statistical analysis has been  supplemented in this section (line 219,225,230,241,251,256,318,319,327,328,343,344,350,355,356,371).

  1. In the section introducing the classification of nanocarriers (2. Categories for Nanocarriers), the descriptions of the synthesis methods and chemical structures of various nanocarriers are relatively brief. For instance, the synthesis principles and structural characteristics of commonly used polymers such as PLA, PGA, and PLGA are not further elaborated. Only their applications in drug delivery are mentioned, which may make it difficult for readers unfamiliar with this field to deeply understand the characteristics and advantages of different nanocarriers.

Response: We thank the reviewer for this point and the descriptions of the synthesis methods and chemical structures of various nanocarriers has been added in this section ( line 58 to 185).

  1. Please supplement content related to the synthesis details and structural properties of various nanocarriers. You can cite some classic research literature on preparation methods and illustrate with simple chemical structure diagrams. When describing the application of each nanocarrier in cardiovascular diseases, further explore the relationship between its action mechanism and the structure and properties of the nanocarrier, that is, why a specific nanocarrier is suitable for a specific treatment target.

Response: We thank the reviewer for this point. The synthesis details and structural properties have been incorporated as previously mentioned, and several seminal research articles on preparation methods have been cited. Various types of nanocarriers have been employed in the treatment of cardiovascular diseases; however, no specific nanocarrier has yet been identified as uniquely suitable for a particular therapeutic target within this domain. We think that Figure 1 may be misleading and have therefore renamed it to "Examples of Gene Therapy Based on Nanocarriers in Cardiovascular Diseases."

  1. The author can add more detailed analyses of actual clinical cases, including the impact of individual patient differences (such as age, gender, comorbidities, etc.) on treatment outcomes. Meanwhile, introduce analyses of some failure cases to explore possible causes of failure, such as the immunogenicity of nanocarriers and the instability of drug release, to make the analysis more comprehensive.

Response: We thank the reviewer for this point. Clinical applications of gene therapy utilizing nanocarriers in cardiovascular contexts remain limited. Currently, clinical efforts in this area primarily focus on targeting PCSK9, and we have provided additional details on this aspect (lines 275 to 299). Previous studies have demonstrated satisfactory safety and efficacy, although some adverse events, such as rash, have been observed following the administration of lipid-based nanocarriers. Strategies to mitigate this immunogenicity, including pre-treatment with antihistamines and dexamethasone, have been employed. However, due to the small scale of early clinical trials, further studies are required to obtain sufficient data for statistical analysis.

  1. The article mentions the clinical trial progress of some gene therapy products, such as VERVE - 101, etc. However, the discussion of the key clinical information of these trials is not comprehensive enough. For example, in the section on hypertension treatment, the nanocarrier gene therapy is not comprehensively compared with traditional thiazide diuretics, angiotensin - converting enzyme (ACE) inhibitors, and calcium channel blockers (CCBs), limiting the guiding value of the article for clinical practice.

Response: We thank the reviewer for this point. We have added this comparison in line 475 to 477. High specificity and more sustained therapeutic effects are the main advantages of nanocarrier-based gene therapy, compared to the traditional drugs.    

  1. On the existing basis, add multiple types of diagrams. For example, draw schematic diagrams of the structures of nanocarriers to show their action processes within cells; create diagrams that associate disease pathological processes with the treatment targets of nanocarriers to more clearly present the treatment mechanisms.

Response: We thank the reviewer for this suggestion. Schematic diagrams of the synthesis process and structures of nanocarriers (Figure 3) and their action processes within cells (Figure 2) have been provided in this manuscript. Furthermore, pathological processes with the treatment targets of nanocarriers in atherosclerosis and pulmonary hypertension have also been provided (Figure 4 and Figure 5, respectively).

  1. It is recommended to use bar charts or line charts to compare the treatment effects of different nanocarriers in cardiovascular diseases, visually demonstrating data differences.

Response: We thank the reviewer for this suggestion. However, due to the heterogeneity present in the studies discussed in our review—encompassing variations in therapeutic cargoes, animal models, therapeutic courses, and outcome indicators—it is challenging to construct bar or line charts directly. Consequently, we have compiled these studies in Table 1, facilitating a more accessible comparison of their details for the readers.

  1. There are no relevant references cited regarding the water solubility and first - pass effect of antihypertensive drugs in lines 40 - 42.

Response: We thank the reviewer for this point and the reference has been added.

  1. The article is clearly structured, proceeding in sequence according to the classification of nanocarriers (2. Categories for Nanocarriers), their applications in different cardiovascular diseases (Sections 3 - 6 discuss different diseases respectively), and challenges and prospects (7. Challenges and Outlook). However, in the discussion of each disease, the pathological processes and corresponding treatment strategies are further subdivided. For the complex classification of nanocarriers, action mechanisms, and disease pathological processes, the content in the article may not be clearly explained. Consider adding more detailed diagrams to help readers better understand.

Response: We thank the reviewer for this suggestion and diagrams that associate disease pathological processes with the treatment targets of nanocarriers in atherosclerosis (Figure 4) and pulmonary hypertension (Figure 5) have been added in this manuscript.

11.The expressions of some professional terms in the article are inconsistent, affecting readers' understanding of the content. Standardize the use of professional terms throughout the article. Provide a complete definition and explanation when a term first appears and maintain consistency thereafter.

Response: We thank the reviewer for this suggestion and the professional terms used in this manuscript have been further standardized.

  1. Some sentences have complex structures and obscure expressions, reducing the readability of the article. Please simplify and optimize complex sentences.

Response: We thank the reviewer for this suggestion and we are going to get the professional English language editing from MDPI Author Services for help, if it is necessary.

Reviewer 3 Report

Comments and Suggestions for Authors

This paper provides a concise and informative overview of cardiovascular diseases and the potential of nanocarrier-based gene therapy as a promising alternative. It clearly outlines the scope, focusing on various cardiovascular conditions and emphasizing the novelty of addressing valvular heart disease. However, several aspects could be improved to enhance clarity, impact, and scientific rigor.

1. Varying type of cells are discussed as targets. However, it is not clear why the nanoparticles can target these cells. The authors should discuss. Including a table for clear summary may be better.

2. How do the nanoparticles accumulate in these cardiovascular diseases sites?

3. "Vectors utilized in gene therapy are categorized into viral and non-viral types. The predominant viral vectors encompass adenoviruses (AdVs), adeno-associated viruses (AAVs), retroviruses, and lentiviruses. In contrast, non-viral vectors primarily include techniques such as electroporation, gene guns, fluid pressure, sonoporation, magnetic transfection, liposomes, cell-penetrating peptides, and nanoparticles [14]." electroporation, gene guns, fluid pressure, sonoporation, magnetic transfection can not be treated as non-viral vectors. These are physical strategies.

4. In the conclusion, the authors discuss the issue of the accumulation in the liver. PEGylation stands out among the techniques explored so far to decrease the liver accumulation (stealth effect), accelerating the timeline of nanocarriers towards clinical translation across multiple types, including PEGylated protein, lipid-, polymer-, and inorganic nanoparticle-based systems (https://doi.org/10.1016/j.addr.2023.114895). The optimal stealth effect of nanocarriers should largely depend on a specific biomedical application with the careful consideration of the biology/pathology of specific diseases and the type of therapeutic cargo. Successful examples of nanocarriers should keep a good balance between stealth effect and interaction with diseased tissue. Moving forward, to break the trade-off between these two situations, dynamic modulation of the stealth effect through stimuli-responsive strategy may further advance the therapeutic efficacy. The authors should provide more insights on this point.

5. The authors should discuss more on the design of nanocarriers, such as improving functionality for targeting cardiovascular diseases (https://doi.org/10.1021/jacs.0c09029), which is missing significantly.

Author Response

  1. Varying type of cells are discussed as targets. However, it is not clear why the nanoparticles can target these cells. The authors should discuss. Including a table for clear summary may be better.

Response: We thank the reviewer for this point. The mechanisms of the accumulation of nano carriers in these cardiovascular diseases sites include passive-targeting and active targeting. Passive targeting is widely used in oncology because of the "enhanced permeation and retention" (EPR) effect, which is notably significant in tumors. This effect, promoting nanoparticle accumulation, is influenced by vascular factors involved in inflammation. As a result, the EPR effect can also be observed in other inflammatory conditions, including arthritis, infections, and advanced atherosclerotic plaques. Active targeting employs affinity ligands to bind nanoparticles to antigens or extracellular matrix proteins that are overexpressed on diseased cell membranes or tissues and active targeting nanoparticles primarily utilize ligands such as monoclonal antibodies and their fragments, proteins or protein-like molecules, peptides, nucleic acid ligands, small molecules, and sugars. Our review highlights that monoclonal antibodies, peptides, and small molecules are frequently used for active targeting nanoparticles in cardiovascular diseases. 

  We have added detailed discussion of active targeting in line 621-638 and summarized the active strategy of these nanocarriers in Table 1.

  1. How do the nanoparticles accumulate in these cardiovascular diseases sites?

Response: We thank the reviewer for this point. As mentioned above, the mechanisms include passive-targeting and active targeting, and we have added detailed discussion of active targeting in line 621-638 and summarized the active strategy of these nanocarriers in Table 1.  

  1. "Vectors utilized in gene therapy are categorized into viral and non-viral types. The predominant viral vectors encompass adenoviruses (AdVs), adeno-associated viruses (AAVs), retroviruses, and lentiviruses. In contrast, non-viral vectors primarily include techniques such as electroporation, gene guns, fluid pressure, sonoporation, magnetic transfection, liposomes, cell-penetrating peptides, and nanoparticles [14]." electroporation, gene guns, fluid pressure, sonoporation, magnetic transfection can not be treated as non-viral vectors. These are physical strategies.

Response: We thank the reviewer for this point and we have corrected our statement in the manuscript.

  1. In the conclusion, the authors discuss the issue of the accumulation in the liver. PEGylation stands out among the techniques explored so far to decrease the liver accumulation (stealth effect), accelerating the timeline of nanocarriers towards clinical translation across multiple types, including PEGylated protein, lipid-, polymer-, and inorganic nanoparticle-based systems (https://doi.org/10.1016/j.addr.2023.114895). The optimal stealth effect of nanocarriers should largely depend on a specific biomedical application with the careful consideration of the biology/pathology of specific diseases and the type of therapeutic cargo. Successful examples of nanocarriers should keep a good balance between stealth effect and interaction with diseased tissue. Moving forward, to break the trade-off between these two situations, dynamic modulation of the stealth effect through stimuli-responsive strategy may further advance the therapeutic efficacy. The authors should provide more insights on this point.

Response: We thank the reviewer for this point and a more detailed discussion have been added to this section ( line 570-593).

  1. The authors should discuss more on the design of nanocarriers, such as improving functionality for targeting cardiovascular diseases (https://doi.org/10.1021/jacs.0c09029), which is missing significantly.

Response: We thank the reviewer for this suggestion and a discussion on the design of active-targeting nanocarriers have been added in this part (line 621-638).

Round 2

Reviewer 1 Report

Comments and Suggestions for Authors

Review the spaces and dots before and after cites in the text. 

Reviewer 2 Report

Comments and Suggestions for Authors

This paper can be accepted

Reviewer 3 Report

Comments and Suggestions for Authors

The authors have now addressed all my concerns. Table is very helpful. It is very good informative paper!